# Corruption-Robust Offline Reinforcement Learning with General Function Approximation

**Chenlu Ye**[*]
The Hong Kong University of Science and Technology
cyeab@connect.ust.hk

**Rui Yang**[*]
The Hong Kong University of Science and Technology
ryangam@connect.ust.hk

**Quanquan Gu**
University of California, Los Angeles
qgu@cs.ucla.edu

**Tong Zhang**
The Hong Kong University of Science and Technology
tongzhang@ust.hk

## Abstract

We investigate the problem of corruption robustness in offline reinforcement learning (RL) with general function approximation, where an adversary can corrupt each sample in the offline dataset, and the corruption level $\zeta \geq 0$ quantifies the cumulative corruption amount over $n$ episodes and $H$ steps. Our goal is to find a policy that is robust to such corruption and minimizes the suboptimality gap with respect to the optimal policy for the uncorrupted Markov decision processes (MDPs). Drawing inspiration from the uncertainty-weighting technique from the robust online RL setting [18, 55], we design a new uncertainty weight iteration procedure to efficiently compute on batched samples and propose a corruption-robust algorithm for offline RL. Notably, under the assumption of single policy coverage and the knowledge of $\zeta$, our proposed algorithm achieves a suboptimality bound that is worsened by an additive factor of $\mathcal{O}(\zeta \cdot (\mathrm{CC}(\lambda, \widehat{\mathcal{F}}, \mathcal{Z}_n^H))^{1/2}(C(\widehat{\mathcal{F}}, \mu))^{-1/2}n^{-1})$ due to the corruption. Here $\mathrm{CC}(\lambda, \widehat{\mathcal{F}}, \mathcal{Z}_n^H)$ is the coverage coefficient that depends on the regularization parameter $\lambda$, the confidence set $\widehat{\mathcal{F}}$, and the dataset $\mathcal{Z}_n^H$, and $C(\widehat{\mathcal{F}}, \mu)$ is a coefficient that depends on $\widehat{\mathcal{F}}$ and the underlying data distribution $\mu$. When specialized to linear MDPs, the corruption-dependent error term reduces to $\mathcal{O}(\zeta d n^{-1})$ with $d$ being the dimension of the feature map, which matches the existing lower bound for corrupted linear MDPs. This suggests that our analysis is tight in terms of the corruption-dependent term.

## 1 Introduction

Offline reinforcement learning (RL) has received tremendous attention recently because it can tackle the limitations of online RL in real-world applications, e.g., healthcare [45] and autonomous driving [35], where collecting online data is risky, expensive and even infeasible. By leveraging a batch of pre-collected datasets, offline RL aims to find the optimal policy that is covered by the dataset without further interaction with the environment. Due to the restriction of the offline dataset, the utilization of pessimism in the face of uncertainty is widespread [1, 2, 53, 40, 16] and plays a central role in

---

[*]These authors contributed equally to this work.

37th Conference on Neural Information Processing Systems (NeurIPS 2023).

providing theoretical guarantees for efficient learning [24, 36, 44, 48, 51, 56, 57, 63, 52]. Notably, these theoretical works demonstrate that a single-policy coverage is sufficient to guarantee sample efficiency.

In this paper, we study offline RL under adversarial corruption and with general function approximation. Adversarial corruption refers to adversarial attacks on the reward functions and transition dynamics on the data at each step before the learner can access the dataset. The learner only knows the cumulative corruption level and cannot tell whether the corruption occurs at each data point. Our corruption formulation subsumes the model misspecification [22] and the fixed fraction of data contamination [60] as special cases. Various real-world problems are under the threat of adversarial corruption, such as chatbots misled by discriminative or unethical conversations [32, 61], and autonomous vehicles tricked by hacked navigation instructions or deliberately contaminated traffic signs [12]. On the other hand, general function approximation (approximating the value function with a nonlinear function class, such as deep neural networks) plays a pivotal role in modern large-scale RL problems, such as large language model [10], robotics [26] and medical treatment [28]. Recent works have established different frameworks to explore the minimal structure condition for the function class that enables sample efficiency [19, 47, 9, 21, 14, 6, 64]. In particular, Wang et al. [47] leverage the concept of eluder dimension [39] and construct the least squares value iteration (LSVI) framework, which establishes optimism at each step for online RL.

For adversarial corruption and general function approximation, a significant amount of research has focused on the online setting. However, offline RL in this setting is still understudied due to restricted coverage conditions and lack of adaptivity. One notable exception is Zhang et al. [60], which assumes $\epsilon$-fraction of the offline dataset is corrupted, and their algorithm suffers from a suboptimal bound on the corruption term. Our work moves a step further and achieves corruption robustness under the LSVI framework [47] in offline RL with general function approximation by generalizing the uncertainty weighting technique [18, 55]. We also propose an algorithm robust to an additional known distribution shift. Due to space limit, we defer it to Appendix C. We summarize our contributions as follows:

- We formally define the corruption level in offline RL. With knowledge of the corruption level, we design an algorithm that draws inspiration from the PEVI algorithm [24] and the uncertainty-weighting technique. The uncertainty for each data point, serving as the bonus function, is quantified by its informativeness with respect to the whole dataset. We propose the uncertainty weight iteration algorithm to calculate the weights efficiently and prove that the output, an approximation of the uncertainty, is sufficient to control the corruption term.

- Theoretically, our proposed algorithm enjoys a suboptimality bound of $\tilde{\mathcal{O}}(H \cdot \mathrm{CC}^{1/4}(\ln N)^{1/2}(C(\widehat{\mathcal{F}}, \mu))^{-1/4}n^{-1/2} + \zeta \cdot \mathrm{CC}^{1/2}(C(\widehat{\mathcal{F}}, \mu))^{-1/2}n^{-1})$, where $H$ is the episode length, $n$ is the number of episodes, $N$ is the covering number, CC is the coverage coefficient, and $C(\widehat{\mathcal{F}}, \mu)$ is the coefficient depicting how well the underlying data distribution $\mu$ explores the feature space, and $\widehat{\mathcal{F}}$ is the confidence set. The corruption-dependent term reduces to $\mathcal{O}(\zeta dn^{-1})$ in the linear model of dimension $d$, thus matching the lower bound for corrupted linear MDPs. It is worth highlighting that our novel analysis enables us to eliminate the uncertainty-related weights from the coverage condition.

- Motivated by our theoretical findings, we present a practical offline RL algorithm with uncertainty weighting and demonstrate its efficacy under diverse data corruption scenarios. Our practical implementation achieves a $104\%$ improvement over the previous state-of-the-art uncertainty-based offline RL algorithm under data corruption, demonstrating its potential for effective deployment in real-world applications.

## 1.1 Related Work

**Corruption-Robust Bandits and RL.** There is an emerging body of theoretical literature on bandits and online RL with corruption. The adversarial corruption is first formulated in the multi-armed bandit problem by Lykouris et al. [30], where an adversary corrupts the reward in each round $t$ by $\zeta_t$ and the corruption level is measured by $\zeta = \sum_{t=1}^{T} |\zeta_t|$. Then, a lower bound with a linear dependence on $\zeta$ is constructed by Gupta et al. [17], indicating that the ideal regret bound should achieve a "parallel" relationship: $\mathrm{Regret}(T) = o(T) + \mathcal{O}(\zeta)$, and the corruption-independent term approximates the non-corrupted bound. When extending to linear contextual bandits, a line of

work [4, 8, 13, 27, 62, 25] propose various methods but either derive sub-optimal regrets or require particular assumptions. The gap is later closed by He et al. [18], which achieves the minimax lower bound using a novel sample-dependent weighting technique. Specifically, the weight for each sample is adaptive to its confidence, which is also called uncertainty. Beyond bandits, earlier works on MDPs [5, 20, 23, 29, 33, 37, 38] consider the setting where only the rewards are corrupted, and the transitions remain intact. Wu et al. [50] begin to handle corruption on both rewards and transitions for tabular MDPs. Wei et al. [48] establish a unified framework for RL with unknown corruption under a weak adversary, where the corruption happens before the decision is made in each round. Later, Ye et al. [55] extend the weighting technique [18] to corrupted RL with general function approximation and achieve a linear dependence on the cumulative corruption level $\zeta$. Particularly, Wei et al. [48], Ye et al. [55] both impose corruption on the Bellman operator, which is the same as the corruption model considered in this paper.

**Offline RL Against Attacks.** The emergence of poisoning attacks in real-world scenarios poses new challenges for offline RL and necessitates improved defenses [49]. There are generally two types of attacks [3], namely test-time attacks and training-time attacks. In test-time attacks, the training data is clean, and the learned policy must contend with an attacker during test time. For example, Yang et al. [53] propose learning conservative and smooth policies robust to different test-time attacks. In contrast, our paper focuses on the training-time attack as another line of work [31, 49, 60], where part of the training data is corrupted maliciously. Wu et al. [49] propose two certification criteria and a new aggregation-based method to improve the learned policy from corrupted data. To the best of our knowledge, [60] is the only theoretical work on corrupted offline RL, which considers that an $\epsilon$-fraction ($\epsilon = \zeta/nH$) of samples are corrupted on both rewards and transitions for linear MDPs and achieves an $\mathcal{O}(\zeta^{1/2}dn^{-1/2})$ suboptimality bound. Notably, distinct from the setting in Zhang et al. [60] that clean data is first collected and then corrupted by an adversary, we consider the setting that data collection and corruption occur at the same time (thus corruption at one step affects the subsequent trajectory). Therefore, our setting is different from that of Zhang et al. [60].

## 2 Preliminaries

In this section, we formulate the episodic Markov decision process (MDP) with adversarial corruption and under general (nonlinear) function approximation. Before the formal introduction, we introduce some notations to facilitate our presentation.

**Notations.** Let $[n]$ denote the set $\{1, \ldots, n\}$. For spaces $\mathcal{X}$ and $\mathcal{A}$ and a function $f : \mathcal{X} \times \mathcal{A} \to \mathbb{R}$, let $f(x) = \max_{a \in \mathcal{A}} f(x, a)$. Given a semi-definite matrix $M$ and a vector $v$, we define $\|v\|_M = \sqrt{v^\top M v}$. For two positive sequences $\{f(n)\}_{n=1}^{\infty}$, $\{g(n)\}_{n=1}^{\infty}$, let $f(n) = \mathcal{O}(g(n))$ if there exists a constant $C > 0$ such that $f(n) \leq Cg(n)$ for all $n \geq 1$, and $f(n) = \Omega(g(n))$ if there exists a constant $C > 0$ such that $f(n) \geq Cg(n)$ for all $n \geq 1$. We use $\tilde{\mathcal{O}}(\cdot)$ to omit polylogarithmic factors. Sometimes we use the shorthand notation $z = (x, a)$.

### 2.1 Episodic MDPs

We consider an episodic MDP $(\mathcal{X}, \mathcal{A}, H, \mathbb{P}, r)$ with the state space $\mathcal{X}$, action space $\mathcal{A}$, episode length $H$, transition kernel $\mathbb{P} = \{\mathbb{P}^h\}_{h \in [H]}$, and reward function $r = \{r^h\}_{h \in [H]}$. Suppose that the rewards are bounded: $r^h \geq 0$ for any $h \in [H]$, and $\sum_{h=1}^{H} r^h(x^h, a^h) \leq 1$ almost surely. Given any policy $\pi = \{\pi^h : \mathcal{X} \to \mathcal{A}\}_{h \in [H]}$, we define the Q-value and V-value functions starting from step $h$ as

$$Q_\pi^h(x^h, a^h) = \sum_{h'=h}^{H} \mathbb{E}_\pi \big[ r^{h'}(x^{h'}, a^{h'}) \,|\, x^h, a^h \big], \quad V_\pi^h(x^h) = \sum_{h'=h}^{H} \mathbb{E}_\pi \big[ r^{h'}(x^{h'}, a^{h'}) \,|\, x^h \big]. \quad (1)$$

where the expectation $\mathbb{E}_\pi$ is taken with respect to the trajectory under the policy $\pi$. There exists an optimal policy $\pi_*$ and optimal value functions $V_*^h(x) := V_{\pi^*}^h(x) = \sup_\pi V_\pi^h(x)$ and $Q_*^h(x, a) := Q_{\pi^*}^h(x, a) = \sup_\pi Q_\pi^h(x, a)$ that satisfy the Bellman optimality equation:

$$Q_*^h(x, a) = \mathbb{E}_{r^h, x^{h+1}} \big[ r^h(s, a) + \max_{a' \in \mathcal{A}} Q_*^{h+1}(x^{h+1}, a') \,|\, x, a \big] := (\mathcal{T}^h Q_*^{h+1})(x, a), \quad (2)$$

where $\mathcal{T}^h$ is called the Bellman operator. Then we define the Bellman residual as

$$\mathcal{E}^h(f, x^h, a^h) = f^h(x^h, a^h) - (\mathcal{T}^h f^{h+1})(x^h, a^h). \quad (3)$$

## 2.2 General Function Approximation

We approximate the Q-value functions by a function class $\mathcal{F} = \mathcal{F}_1 \times \cdots \times \mathcal{F}_H$ where $\mathcal{F}_h : \mathcal{X} \times \mathcal{A} \to [0, 1]$ for $h \in [H]$, and $f_{H+1} \equiv 0$ since no reward is generated at step $H + 1$. Generally, the following assumption is common for the approximation function class.

**Assumption 2.1** (Realizability and Completeness). *For all $h \in [H]$, $Q_*^h \in \mathcal{F}^h$. Additionally, for all $g^{h+1}(x^{h+1}) \in [0, 1]$, $(\mathcal{T}^h g^{h+1})(x^h, a^h) \in \mathcal{F}^h$.*

The realizability assumption [21] ensures the possibility of learning the true Q-value function by considering the function class $\mathcal{F}$. The Bellman completeness (adopted from Wang et al. [47] and Assumption 18.22 of Zhang [59]) is stronger than that in Jin et al. [21]. The former applies the least squares value iteration (LSVI) algorithm that establishes optimism at each step, while the latter proposes the GOLF algorithm that only establishes optimism at the first step. We use the standard covering number to depict the scale of the function class $\mathcal{F}$.

**Definition 2.1** ($\epsilon$-Covering Number). *The $\epsilon$-covering number $N(\epsilon, \mathcal{F}, \rho)$ of a set $\mathcal{F}$ under metric $\rho$ is the smallest cardinality of a subset $\mathcal{F}_0 \subseteq \mathcal{F}$ such that for any $f \in \mathcal{F}$, there exists a $g \in \mathcal{F}_0$ satisfying that $\rho(f, g) \leq \epsilon$. We say $\mathcal{F}_0$ is an $(\epsilon, \rho)$ cover of $\mathcal{F}$.*

## 2.3 Offline Data Collection Process

**Offline Clean Data.** Consider an offline clean dataset with $n$ trajectories $\mathcal{D} = \{(x_i^h, a_i^h, r_i^h)\}_{i,h=1}^{n,H}$. We assume the dataset $\mathcal{D}$ is compliant with an MDP $(\mathcal{X}, \mathcal{A}, H, \mathbb{P}, r)$ with the value functions $Q, V$ and the Bellman operator $\mathcal{T}$: for any policy $\pi$,

$$\mathbb{P}\big((\mathcal{T}^h Q_\pi^{h+1})(x_i^h, a_i^h) = r' + V_\pi^{h+1}(x')\big|\{(x_j^h, a_j^h)\}_{j \in [i]}, \{r_j^h, x_j^{h+1}\}_{j \in [i-1]}, Q_\pi^{h+1}\big)$$
$$= \mathbb{P}\big(r^h(x^h, a^h) = r', x^{h+1} = x'\big|x^h = x_i^h, a^h = a_i^h\big), \quad (4)$$

where the realizability and completeness in Assumption 2.1 hold. The compliance assumption (4) is also made in Jin et al. [20], Zhong et al. [63], which means that $\mathcal{D}$ remains the Markov property and allows $\mathcal{D}$ to be collected by an adaptive behavior policy. The induced distribution of the state-action pair is denoted by $\mu = \{\mu^h\}_{h \in [H]}$.

**Adversarial Corruption.** During the offline dataset collection process, after observing the state-action pair $(x^h, a^h)$ chosen by the data collector, an adversary corrupts $r^h$ and $x^{h+1}$ at each step $h$ before they are revealed to the collector. For the corrupted dataset $\mathcal{D}$, we define the corrupted value function $Q_\mathcal{D}$, $V_\mathcal{D}$, and the Bellman operator $\mathcal{T}_\mathcal{D}$ satisfying (4). To measure the corruption level, we notice that characterizing the specific modification on each tuple $(s, a, s', r)$ is hard and unnecessary since once one modifies a tuple, the subsequent trajectory changes. Therefore, it is difficult to tell whether the change is caused by the corruption at the current step or a previous step. In fact, we only care about the part of the change that violates the Bellman completeness. Therefore, following the online setting [55, 48], we measure the corruption level by the gap between $\mathcal{T}_\mathcal{D}$ and $\mathcal{T}$ as follows.

**Definition 2.2** (Cumulative Corruption). *The cumulative corruption is $\zeta$ if at any step $h \in [H]$, for a sequence $\{(x_i^h, a_i^h)\}_{i,h=1}^{n,H} \subset \mathcal{X} \times \mathcal{A}$ chosen by the data collector and a sequence of functions $\{g^h : \mathcal{X} \to [0, 1]\}_{h=1}^H$, we have for all $h \in [H]$,*

$$\sum_{i=1}^n |\zeta_i^h| \leq \zeta^h, \quad \sum_{h=1}^H \zeta^h := \zeta,$$

*where $\zeta_i^h = (\mathcal{T}^h g^{h+1} - \mathcal{T}_\mathcal{D}^h g^{h+1})(x_i^h, a_i^h)$.*

The **learning objective** is to find a policy $\pi$ that minimizes the suboptimality of $\pi$ given any initial state $x^1 = x$: $\text{SubOpt}(\pi, x) = V_*^1(x) - V_\pi^1(x)$, where $V(\cdot)$ is the value function induced by the uncorrupted MDP.

# 3 Algorithm

In this section, we first highlight the pivotal role that uncertainty weighting plays in controlling the corruption-related bound. To extend the uncertainty weighting technique to the offline setting, we propose an iteration algorithm. With the proposed algorithm, the theoretical result for the suboptimality is presented.

## 3.1 Uncertainty-Related Weights

In this subsection, we discuss the choice of weight for a simplified model without state transition ($H = 1$) and use the notation $z_i = (x_i, a_i)$. Given a dataset $\{(z_i, y_i)\}_{i \in [n]}$, we have $y_i = \bar{f}(z_i) + \zeta_i + \epsilon_i$ for $i \in [n]$, where $\bar{f} \in \mathcal{F}$ is the uncorrupted true value, the noise $\epsilon_i$ is zero-mean and conditional $\eta$-subGaussian, and the corruption level is $\zeta = \sum_{i=1}^{n} |\zeta_i|$.

We begin with delineating the consequence caused by the adversarial corruption for the traditional least-square regression: $\hat{f} = \min_{f \in \mathcal{F}} \sum_{i=1}^{n} (f(z_i) - y_i)^2$. Some calculations lead to the following decomposition:

$$\sum_{i=1}^{n} (\hat{f}(z_i) - \bar{f}(z_i))^2 = \underbrace{\sum_{i=1}^{n} \left[ (\hat{f}(z_i) - y_i)^2 - (\bar{f}(z_i) - y_i)^2 \right]}_{I_1 \leq 0} + 2 \underbrace{\sum_{i=1}^{n} (\hat{f}(z_i) - \bar{f}(z_i)) \epsilon_i}_{I_2 : \text{Noise term}} + 2 \underbrace{\sum_{i=1}^{n} (\hat{f}(z_i) - \bar{f}(z_i)) \zeta_i}_{I_3 : \text{Corruption term}}.$$

The term $I_1 \leq 0$ since $\hat{f}$ is the solution to the least-square regression. The term $I_2$ is bounded by $\tilde{\mathcal{O}}(\ln N)$ because of the $\eta$-subGaussainity of $\epsilon_i$, where $N$ is the covering number of $\mathcal{F}$. The term $I_3$ is ruined by corruption: $I_3 \leq 2 \sum_{i=1}^{n} |\zeta_i| = \mathcal{O}(\zeta)$. Hence, the confidence radius $(\sum_{i=1}^{n} (\hat{f}(z_i) - \bar{f}(z_i))^2)^{1/2} = \tilde{\mathcal{O}}(\sqrt{\zeta + \ln N})$ will explode whenever the corruption level $\zeta$ grows with $n$.

To control the corruption term, motivated by the uncertainty-weighting technique from online settings [55, 18, 65], we apply the weighted regression: $\hat{f} = \min_{f \in \mathcal{F}} \sum_{i=1}^{n} (f(z_i) - y_i)^2 / \sigma_i^2$, where ideally, we desire the following uncertainty-related weights:

$$\sigma_i^2 = \max \left( 1, \frac{1}{\alpha} \underbrace{\sup_{f, f' \in \mathcal{F}} \frac{|f(z_i) - f'(z_i)|}{\sqrt{\lambda + \sum_{j=1}^{n} (f(z_j) - f'(z_j))^2 / \sigma_j^2}}}_{\text{Uncertainty}} \right), \quad i = 1, \ldots, n, \qquad (5)$$

where $\alpha, \lambda > 0$ are pre-determined parameters. The uncertainty quantity in the above equation is the supremum of the ratio between the prediction error $|f(z_i) - f'(z_i)|$ and the training error $\sqrt{\sum_{j=1}^{n} (f(z_j) - f'(z_j))^2 / \sigma_j^2}$ over $f, f' \in \mathcal{F}$. Intuitively, the quantity depicts the relative information of a sample $z_i$ against the whole training set $\{z_1, \ldots, z_n\}$. We can use the linear function class as a special example to explain it. When the function space $\mathcal{F}^h$ is embedded into a $d$-dimensional vector space: $\mathcal{F}^h = \{\langle w(f), \phi(\cdot) \rangle : z \to \mathcal{R}\}$, the uncertainty quantity becomes

$$\sup_{f, f' \in \mathcal{F}} \frac{|\langle w(f) - w(f'), \phi(z_i) \rangle|}{\sqrt{\lambda + \sum_{j=1}^{n} (\langle w(f) - w(f'), \phi(z_j) \rangle)^2 / \sigma_j^2}} \leq \sup_{f, f' \in \mathcal{F}} \frac{|\langle w(f) - w(f'), \phi(z_i) \rangle|}{\sqrt{(w(f) - w(f'))^\top \Lambda (w(f) - w(f'))}}$$

$$\leq \sqrt{\phi^\top(z_i) \Lambda^{-1} \phi(z_i)},$$

where $\Lambda = \sum_{j=1}^{n} \phi(z_j) \phi^\top(z_j) / \sigma_j^2$. Moreover, $(\phi^\top(z_i) \Lambda^{-1} \phi(z_i))^{-1}$ represents the effective number of samples in the $\{z_i\}_{i=1}^{n}$ along the $\phi(z_i)$'s direction. We discuss in Lemma B.3 that under mild conditions the linear and nonlinear uncertainty quantities are almost equivalent.

However, since the uncertainty also depends on weights, it is impossible to determine all the weights $\{\sigma_i\}_{i \in [n]}$ simultaneously. Compared with the online setting where the weight in each round can be determined sequentially (iteratively in rounds), we face two challenges in the offline setting: (a) how to compute uncertainty-related weights iteratively? (b) will an approximate solution to the uncertainty play an equivalent role in controlling the corruption term?

To solve the first challenge, we propose the weight iteration algorithm in Algorithm 1. Moreover, we demonstrate the convergence of this algorithm by the monotone convergence theorem in the following lemma, which ensures that the output weights are sufficiently close to desired ones (5). The proof is provided in Appendix B.1.

**Lemma 3.1.** *There exists a $T$ such that the output of Algorithm 1 $\{\sigma_i := \sigma_i^{T+1}\}_{i=1}^{n}$ satisfy:*

$$\sigma_i^2 \geq \max \left( 1, \psi(z_i)/2 \right), \quad \sigma_i^2 \leq \max \left( 1, \psi(z_i) \right), \qquad (6)$$

*where $\psi(z_i) = \sup_{f, f' \in \mathcal{F}} \frac{|f(z_i) - f'(z_i)|/\alpha}{\sqrt{\lambda + \sum_{j=1}^{n} (f(z_j) - f'(z_j))^2 / \sigma_j^2}}$.*

---

**Algorithm 1** Uncertainty Weight Iteration

---

1: **Input:** $\{(x_i, a_i)\}_{i=1}^n, \mathcal{F}, \alpha > 0$
2: **Initialization:** $t = 0$, $\sigma_i^0 = 1$, $i = 1, \ldots, n$
3: **repeat**
4: $\quad t \leftarrow t + 1$
5: $\quad (\sigma_i^t)^2 \leftarrow \max \left( 1, \sup_{f,f' \in \mathcal{F}} \frac{|f(x_i,a_i) - f'(x_i,a_i)|/\alpha}{\sqrt{\lambda + \sum_{j=1}^n (f(x_j,a_j) - f'(x_j,a_j))^2/(\sigma_j^{t-1})^2}} \right)$, $i = 1, \ldots, n$
6: **until** $\max_{i \in [n]} \left( \sigma_i^t / \sigma_i^{t-1} \right)^2 \leq 2$
7: **Output:** $\{\sigma_i^t\}_{i=1}^n$

---

For the second challenge, the weighted version $L_n := \sum_{i=1}^n (\hat{f}(z_i) - \bar{f}(z_i))^2 / \sigma_i^2$ can also be decomposed into three terms correspondingly. We can demonstrate that an approximate choice of weights satisfying (6) is sufficient to control the corruption term as

$$\sum_{i=1}^n \frac{(\hat{f}(z_i) - \bar{f}(z_i))\zeta_i}{\sigma_i^2} = \sum_{i=1}^n \frac{|\hat{f}(z_i) - \bar{f}(z_i)|\zeta_i \cdot \sqrt{\lambda + L_n}}{\sigma_i^2 \sqrt{\lambda + \sum_{j=1}^n (\hat{f}(z_j) - \bar{f}(z_j))^2/\sigma_j^2}} \leq 2\alpha\zeta\sqrt{\lambda + L_n}.$$

Since the corruption-unrelated terms (corresponding to $I_1, I_2$) can still be bounded by $\tilde{\mathcal{O}}(\ln N)$, we have $L_n = \tilde{\mathcal{O}}(\ln N + 2\alpha\zeta\sqrt{L_n})$, leading to an $\tilde{\mathcal{O}}(\alpha\zeta + \sqrt{\ln N})$ confidence radius. Therefore, with a sufficiently small $\alpha$, the effect of corruption can be countered.

### 3.2 Corruption-Robust Algorithm

---

**Algorithm 2** CR-PEVI

---

1: **Input:** $\mathcal{D} = \{(x_i^h, a_i^h, r_i^h)\}_{i,h=1}^{n,H}, \mathcal{F}$
2: **Initialization:** Set $f_n^{H+1}(\cdot) \leftarrow 0$
3: **for** step $h = H, H-1, \ldots, 1$ **do**
4: $\quad$ Choosing weights $\{\sigma_i^h\}_{i=1}^n$ by proceeding Algorithm 1 with inputs $\{(x_i^h, a_i^h)\}_{i=1}^n, \mathcal{F}^h, \alpha$
5: $\quad$ Find the weighted least-squares solution in (7)
6: $\quad$ Find $\beta^h$ and construct confidence set

$$\widehat{\mathcal{F}}^h = \left\{ f \in \mathcal{F}^h : \lambda + \sum_{i=1}^n (f(x_i^h, a_i^h) - \hat{f}^h(x_i^h, a_i^h))^2 / (\sigma_i^h)^2 \leq (\beta^h)^2 \right\}$$

7: $\quad$ Construct bonus function as (8)
8: $\quad$ Let $f_n^h(\cdot, \cdot) = \max \left( 0, \hat{f}^h(\cdot, \cdot) - \beta^h b^h(\cdot, \cdot) \right)$
9: $\quad$ Set $\hat{\pi}^h(\cdot) = \operatorname{argmax}_{a \in \mathcal{A}} f_n^h(\cdot, a)$
10: **end for**
11: **Output:** $\{\hat{\pi}^h\}_{h=1}^H$

---

Now, for the offline RL with general function approximation, we integrate the uncertainty weight iteration algorithm with the pessimistic value iteration (PEVI) algorithm [24], and propose a Corruption-Robust PEVI (CR-PEVI) in Algorithm 2. Our algorithm employs backward induction from step $H$ to 1. Set estimated value function $f_n^{H+1}(\cdot) = 0$. At each step $h \in [H]$, having obtained $f_n^{h+1}$, we calculate $f_n^h$ by solving the following weighted least-square regression:

$$\hat{f}^h = \operatorname*{argmin}_{f^h \in \mathcal{F}^h} \sum_{i=1}^n \frac{(f(x_i^h, a_i^h) - r_i^h - f_n^{h+1}(x_i^{h+1}))^2}{(\sigma_i^h)^2}, \tag{7}$$

where the weights are obtained via Algorithm 1. As opposed to online RL where the necessity of exploration stimulates optimistic estimation, the literature on offline RL [24, 63] is more inclined to pessimism due to the limitation of offline data coverage. Hence, we construct a confidence set $\widehat{\mathcal{F}}^h = \{f \in \widehat{\mathcal{F}}^h : \lambda + \sum_{i=1}^n (f(x_i^h, a_i^h) - \hat{f}^h(x_i^h, a_i^h))^2 / (\sigma_i^h)^2 \leq (\beta^h)^2\}$ such that the uncorrupted Bellman operator $\mathcal{T}^h$ converts the value function $f_n^{h+1}$ into the function class $\widehat{\mathcal{F}}^h$ (i.e., $\mathcal{T}^h f_n^{h+1} \in$

$\widehat{\mathcal{F}}^h$) with high probability. For the bonus function, we follow [55] and choose it as

$$b_h(x,a) = \sup_{f,f' \in \widehat{\mathcal{F}}^h} \frac{|f(x,a) - f'(x,a)|}{\sqrt{\lambda + \sum_{i=1}^n (f(x_i^h, a_i^h) - f'(x_i^h, a_i^h))^2/(\sigma_i^h)^2}}, \tag{8}$$

which is seldom used in practical algorithms due to its instability. Specifically, the covering number of the space containing (8) may be uncontrollable. According to Appendix E in [55], the issue of the covering number can be addressed under mild conditions by some techniques. Therefore, to maintain readability and consistency in this paper, we assume the corresponding bonus function space $\mathcal{B}^{h+1}$ of (8) has a bounded covering number. Then we introduce pessimism by subtracting $b^h$ from the estimated value function: $f_n^h(x,a) = \max(0, \hat{f}^h(x,a) - \beta^h b^h(x,a))$.

## 4 Theoretical Analysis

### 4.1 Coverage Condition

No guarantee for the suboptimality can be provided with insufficient data coverage. Based on pessimism, Jin et al. [20], Rashidinejad et al. [36] have demonstrated that the coverage over the optimal policy is sufficient for sample-efficient offline RL. The following condition covers the optimal policy under general function approximation.

**Definition 4.1** (Coverage Coefficient). *Consider the offline dataset $\{x_i^h, a_i^h\}_{i,h=1}^{n,H}$. For any initial state $x^1 = x \in \mathcal{X}$, the coverage coefficient is:*

$$\text{CC}(\lambda, \widehat{\mathcal{F}}, \mathcal{Z}_n^H) = \max_{h \in [H]} \mathbb{E}_{\pi_*} \left[ \sup_{f,f' \in \widehat{\mathcal{F}}^h} \frac{n(f(x^h, a^h) - f'(x^h, a^h))^2}{\lambda + \sum_{i=1}^n (f(x_i^h, a_i^h) - f'(x_i^h, a_i^h))^2} \,\bigg|\, x^1 = x \right], \tag{9}$$

*where $\mathbb{E}_{\pi_*}$ is taken with respect to the trajectory induced by $\pi_*$ in the underlying uncorrupted MDP.*

In the face of corruption, we require single-policy coverage over the uncorrupted trajectory. This coefficient depicts the expected uncertainty of sample $(x^h, a^h)$ induced by the optimal policy $\pi_*$ compared to the $n$ training samples. We use the linear MDP to interpret this condition, where the function space $\mathcal{F}^h$ is embedded into a $d$-dimensional vector space: $\mathcal{F}^h = \{\langle w(f), \phi(\cdot) \rangle : z \to \mathbb{R}\}$, where $z$ denotes the state-action pair $(x,a)$. With the notation $\Lambda^h = \lambda I + \sum_{i=1}^n \phi(z_i^h)\phi(z_i^h)^\top$, we can demonstrate that if the sufficient "coverage" in Jin et al. [24] holds: there exists a constant $c^\dagger$ such that $\Lambda^h \succeq I + c^\dagger n \mathbb{E}_{\pi_*}[\phi(z^h)\phi(z^h)^\top \mid x^1 = x]$ for all $h \in [H]$, our coverage coefficient is bounded: $\text{CC}(\lambda, \widehat{\mathcal{F}}, \mathcal{Z}_n^H) \leq d/c^\dagger < \infty$. Therefore, the sufficient "coverage" condition implies our coverage condition in the linear setting. We will discuss the connection formally in Lemma B.1.

### 4.2 Main Result

Before presenting the main theorem, we introduce a new general version of the well-explored dataset condition, which is the key to eliminating the uncertainty-related weights from the suboptimality bound and deriving the final result.

**Assumption 4.1** (Well-Explored Dataset). *There exists a constant $C(\widehat{\mathcal{F}}, \mu) > 8n^{-1}$ such that for a $(n^{-1}, \|\cdot\|_\infty)$ cover of $\widehat{\mathcal{F}}^h$, denoted by $\bar{\mathcal{F}}^h$, and for any two distinct $f, f' \in \bar{\mathcal{F}}^h$,*

$$\min_{h \in [H]} \sum_{z^h \in \mathcal{X} \times \mathcal{A}} \mu^h(z^h)\big(f(z^h) - f'(z^h)\big)^2 \geq C(\widehat{\mathcal{F}}, \mu). \tag{10}$$

We interpret this condition with the linear model, where the condition (10) becomes: for any two distinct $f, f' \in \bar{\mathcal{F}}^h$, $\min_{h \in [H]} \big(w(f) - w(f')\big)^\top \bar{\Lambda}^h \big(w(f) - w(f')\big) \geq C(\widehat{\mathcal{F}}, \mu)$. As proved in Lemma B.2, with high probability, the above condition holds with $C(\widehat{\mathcal{F}}, \mu) = \Theta(d^{-1})$ when the $n$ trajectories of $\mathcal{D}$ are independent, and the data distribution $\mu^h$ satisfies the minimum eigenvalue condition:

$$\sigma_{\min}\big(\mathbb{E}_{z^h \sim \mu^h}[\phi(z^h)\phi(z^h)^\top]\big) = \bar{c}/d, \tag{11}$$

where $\bar{c} > 0$ is an absolute constant. This is a widely-adopted assumption in the literature [11, 46, 63]. Note that $\Theta(d^{-1})$ is the largest possible minimum eigenvalue since for any data distribution $\tilde{\mu}^h$, $\sigma_{\min}\big(\mathbb{E}_{z^h \sim \mu^h}[\phi(z^h)\phi(z^h)^\top]\big) \leq d^{-1}$ by using $\|\phi(z^h)\| \leq 1$ for any $z^h \in \mathcal{X} \times \mathcal{A}$.

**Theorem 1.** *If the coverage coefficient in Definition 4.1 is finite, under Assumption 4.1, for corruption* $\zeta = \sum_{h=1}^{H} \zeta^h$ *and* $\delta > 0$*, we choose the covering parameter* $\gamma = 1/(n \max_h \beta^h \zeta^h)$*,* $\lambda = \ln(N_n(\gamma))$*, the weighting parameter* $\alpha = H\sqrt{\ln N_n(\gamma)}/\zeta$*, and the confidence radius*

$$\beta^h = c_\beta\big(\alpha\zeta^h + \sqrt{\ln(HN_n(\gamma/\delta))}\big), \quad for\ h = H, \dots, 1,$$

*where* $N_n(\gamma) = \max_h N(\gamma/n, \mathcal{F}^h) \cdot N(\gamma/n, \mathcal{F}^{h+1}) \cdot N(\gamma/n, \mathcal{B}^{h+1}(\lambda))$*. Then, with probability at least* $1 - 2\delta$*, the sub-optimality of Algorithm 2 is bounded by*

$$\text{SubOpt}(\hat{\pi}, x) = \tilde{\mathcal{O}}\bigg( \frac{H(\text{CC}(\lambda, \widehat{\mathcal{F}}, \mathcal{Z}_n^H))^{1/4} \cdot (\ln N_n(\gamma))^{1/2}}{n^{1/2}(C(\widehat{\mathcal{F}}, \mu))^{1/4}} + \frac{\zeta(\text{CC}(\lambda, \widehat{\mathcal{F}}, \mathcal{Z}_n^H))^{1/2}}{n(C(\widehat{\mathcal{F}}, \mu))^{1/2}} \bigg).$$

When $\zeta = \mathcal{O}(\sqrt{n})$, our algorithm achieves the same order of suboptimality as the uncorrupted case. Whenever $\zeta = o(n)$, our algorithm is sample-efficient. Moreover, when specialized to the linear MDP with dimension $d$, where $\text{CC}(\lambda, \widehat{\mathcal{F}}, \mathcal{Z}_n^H) = \mathcal{O}(d/c^\dagger)$, $C(\widehat{\mathcal{F}}, \mu) = \Theta(d^{-1})$ and $\ln N_n(\gamma) = \tilde{\mathcal{O}}(d^2)$, the suboptimality bound in Theorem 1 becomes $\tilde{\mathcal{O}}(Hd^{3/2}n^{-1/2} + d\zeta n^{-1})$. The corruption-independent term $\tilde{\mathcal{O}}(Hd^{3/2}n^{-1/2})$ matches that of PEVI [24]. The corruption-dependent term nearly matches the lower bound, as will be discussed later.

**Remark 4.1.** *Although the theory requires a known corruption level* $\zeta$*, in the experiments, we treat the uncertainty ratio* $\alpha = O(1/\zeta)$ *as a tuning hyperparameter. The use of independent and identically distributed (i.i.d.) trajectories in our experiments renders the hyperparameter tuning process straightforward and conducive to optimizing the performance. Additionally, we can offer a choice of* $\alpha = \Theta(1/\sqrt{n})$*. This choice finds support in the online setting [55, 18], where this specific choice of* $\alpha$ *ensures that suboptimality remains in the order of uncorrupted error bound, even when* $\zeta = O(\sqrt{n})$*.*

**Proof sketch.** The detailed proof of Theorem 1 is provided in Appendix A. Here we present a brief proof sketch for the suboptimality bound, which is accomplished by three steps: (1) by Lemma A.1, if the uncorrupted Bellman backup $\mathcal{T}^h f_n \in \widehat{\mathcal{F}}^h$ for each $h \in [H]$, we can bound the suboptimality by the sum of the bonus $\sum_{h=1}^{H} \beta^h \mathbb{E}_{\pi_*}[b^h(x^h, a^h) \mid x^1 = x]$; (2) by Lemma A.2, we demonstrate that an approximate uncertainty weight satisfying (6) is the key to bound the weighted Bellman error $(\sum_{i=1}^{n} ((\hat{f}^h - (\mathcal{T}^h f_n^{h+1}))(x_i^h, a_i^h))^2/(\sigma_i^h)^2)^{1/2}$ by $\beta^h = c_\beta(\alpha\zeta^h + \sqrt{\ln(HN_n(\gamma/\delta))})$; and (3) combining the results in the first two steps, we can obtain the suboptimality bounded by:

$$\text{SubOpt}(\hat{\pi}, x) = \tilde{\mathcal{O}}\big(H(\text{CC}^\sigma(\lambda, \widehat{\mathcal{F}}, \mathcal{Z}_n^H))^{1/2} \cdot (\ln N_n(\gamma))^{1/2} \cdot n^{-1/2} + \zeta \cdot \text{CC}^\sigma(\lambda, \widehat{\mathcal{F}}, \mathcal{Z}_n^H) \cdot n^{-1}\big).$$

Here $\text{CC}^\sigma(\lambda, \widehat{\mathcal{F}}, \mathcal{Z}_n^H)$ denotes the weighted coverage coefficient as follows

$$\text{CC}^\sigma(\lambda, \widehat{\mathcal{F}}, \mathcal{Z}_n^H) = \max_{h \in [H]} \mathbb{E}_{\pi_*}\bigg[ \sup_{f, f' \in \widehat{\mathcal{F}}^h} \frac{n(f(x^h, a^h) - f'(x^h, a^h))^2/\sigma^h(x^h, a^h)^2}{\lambda + \sum_{i=1}^{n}(f(x_i^h, a_i^h) - f'(x_i^h, a_i^h))^2/(\sigma_i^h)^2} \bigg| x^1 = x \bigg], \quad (12)$$

where the weight for the trajectory induced by the optimal policy $\pi_*$ is

$$(\sigma^h(x^h, a^h))^2 = \max\bigg( 1, \sup_{f, f' \in \widehat{\mathcal{F}}^h} \frac{|f(x^h, a^h) - f'(x^h, a^h)|/\alpha}{\sqrt{\lambda + \sum_{i=1}^{n}(f(x_i^h, a_i^h) - f'(x_i^h, a_i^h))^2/(\sigma_i^h)^2}} \bigg). \quad (13)$$

To control the weighted coverage coefficient $\text{CC}^\sigma$ by its unweighted counterpart $\text{CC}$, which is a challenging task due to the intricate form of uncertainty-related weights, a novel technique has been employed. The main idea behind this technique is to transform $\text{CC}^\sigma$ into a modified version of $\text{CC}$ multiplied by a term that can be bounded by leveraging a cover of the function class. The connection between $\text{CC}$ and $\text{CC}^\sigma$ is shown in the following lemma.

**Lemma 4.1.** *Under Assumption 4.1 and choose* $\beta^h = C_\beta\sqrt{\ln N}$ *(where* $C_\beta > 0$ *contains the logarithmic terms that are omitted) and* $\lambda$ *given in Theorem 1, we have*

$$\text{CC}^\sigma(\lambda, \widehat{\mathcal{F}}, \mathcal{Z}_n^H) = \tilde{\mathcal{O}}\big((\text{CC}(\lambda, \widehat{\mathcal{F}}, \mathcal{Z}_n^H))^{1/2} \cdot (C(\widehat{\mathcal{F}}, \mu))^{-1/2}\big).$$

Intuitively, because the weights themselves are closely related to the uncertainty, the weighted coverage coefficient $\text{CC}^\sigma$ where the high-uncertain samples are down-weighted can be upper-bounded by the square root of the unweighted coefficient $\text{CC}$. See Appendix B.3 for details.

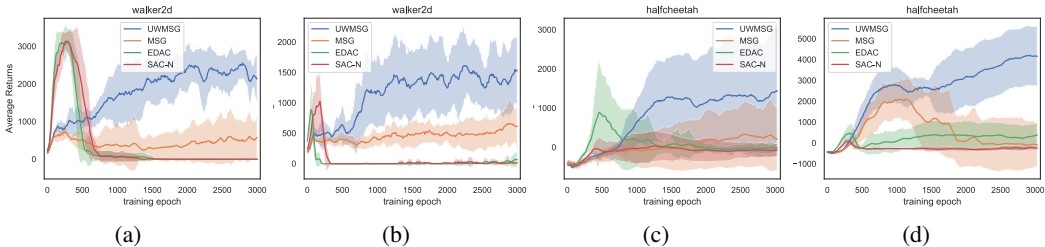

Figure 1: Performance on the Walker2d and the Halfcheetah tasks under (a) random reward, (b) random dynamics, (c) adversarial reward, and (d) adversarial dynamics attacks.

**Lower Bound.** We construct a lower bound for linear MDPs with adversarial corruption $\zeta$ to show that the $\mathcal{O}(d\zeta n^{-1})$ corruption term of our suboptimality is optimal. The construction of the lower bound is adapted from Zhang et al. [60], where an $\epsilon$-constant fraction of the dataset is contaminated. We present the proof of Theorem 2 in Appendix B.4.

**Theorem 2** (Minimax Lower Bound for Linear MDPs). *Under linear MDPs with corruption (Definition 2.2), for any fixed data-collecting distribution $\nu$, any algorithm $L : \mathcal{D} \to \Pi$ with the knowledge of $\zeta$ cannot find a better policy than $\mathcal{O}(d\zeta n^{-1})$-optimal policy with probability more than $1/4$:*

$$\min_{L,\nu} \max_{MDP,\mathcal{D}} \mathrm{SubOpt}(\hat{\pi}_L) = \Omega\big(d\zeta/n\big),$$

*where $\mathcal{D}$ is the corrupted dataset initially generated from the MDP and then corrupted by an adversary, and $\hat{\pi}_L$ is the policy generated by the algorithm L.*

## 5 Experiments

Based on our theoretical results, we propose a practical implementation for CR-PEVI and verify its effectiveness on simulation tasks with corrupted offline data.

**Practical Implementation.** To make our algorithm more practical, we use neural networks to estimate the $Q$ function (i.e., $f$ in our theory) and the weight function $\sigma$. In linear MDPs, under a sufficiently broad function approximation class $\widehat{\mathcal{F}}^h$ and sufficiently small parameter $\lambda$ in Eq. (8), the bonus function can be simplified as the bootstrapped uncertainty of $Q$ functions, which in turn can be estimated via the standard deviation of an ensemble of $Q$ networks. We defer the detailed discussion to Appendix B.5. Following the state-of-the-art uncertainty-based offline RL algorithm Model Standard-deviation Gradients (MSG) [16], we learn a group of $Q$ networks $Q_{w_i}, i = 1, \ldots, K$ with independent targets and optimize a policy $\pi_\theta$ with a lower-confidence bound (LCB) objective [16, 2]. Specifically, $Q_{w_i}$ is learned to minimize a weighted regression objective similar to Eq.(7). The weight function $\sigma$ is estimated via bootstrapped uncertainty: $\sigma(x,a) = \max(\sqrt{\mathbb{V}_{i=1,\ldots,K}[Q_{w_i}(x,a)]}, 1)$, where $\mathbb{V}_{i=1,\ldots,K}[Q_{w_i}]$ is the variance between the group of $Q$ functions. This uncertainty estimation method has also been adopted by prior works [2, 16, 54]. We refer to our practical algorithm as Uncertainty Weighted MSG (UWMSG) and defer details to Appendix D.

**Experimental Setup.** We assess the performance of our approach using continuous control tasks from [15] and introduce both random and adversarial attacks on either the rewards or dynamics for the offline datasets. Details about the four types of data corruption and their cumulative corruption levels are deferred to Appendix D. The ensemble size $K$ is set to 10 for all experiments. For evaluation, we report average returns with standard deviations over 10 random seeds. More implementation details are also provided in Appendix D.

**Experimental Results.** We compare UWMSG with the state-of-the-art uncertainty-base offline RL methods, MSG [16], EDAC [1], and SAC-N [1] under four types of data corruption. In particular, MSG can be considered as UWMSG with a constant weighting function $\sigma(s,a) = 1$. As demonstrated in Table 1 and Figure 1, our empirical results find that (1) current offline RL methods are susceptible to data corruption, e.g., MSG, EDAC, SAC-N achieve poor performance under adversarial attacks, and (2) our proposed UWMSG significantly improves performance under different data corruption scenarios, with an average improvement of $104\%$ over MSG. More results can be found in Appendix F.

Table 1: Comparison of different offline RL algorithms under different environments and different data corruption types.

| Environment | Attack Type | UWMSG | MSG | EDAC | SAC-N |
|---|---|---|---|---|---|
| Halfcheetah | Random Reward | 7299.9 ± 169.0 | 4339.5 ± 3958.7 | 7128.2 ± 120.5 | **7357.5 ± 165.2** |
| | Random Dynamics | **1425.0 ± 1659.5** | 212.9 ± 793.3 | -12.3 ± 131.8 | -66.2 ± 169.9 |
| | Adversarial Reward | **1016.7 ± 503.9** | 243.1 ± 338.6 | -127.2 ± 30.3 | -55.7 ± 24.0 |
| | Adversarial Dynamics | **4144.3 ± 1437.6** | -87.3 ± 1055.6 | 374.0 ± 589.5 | -246.8 ± 104.9 |
| Walker2d | Random Reward | **2189.7 ± 603.0** | 539.1 ± 534.6 | -3.7 ± 1.2 | -3.8 ± 1.4 |
| | Random Dynamics | **2278.9 ± 706.4** | 2122.8 ± 821.5 | -3.6 ± 0.8 | -3.1 ± 0.7 |
| | Adversarial Reward | **1433.4 ± 592.1** | 605.8 ± 310.7 | 61.0 ± 120.6 | 9.5 ± 21.6 |
| | Adversarial Dynamics | **946.0 ± 300.4** | 506.0 ± 175.5 | -4.8 ± 0.3 | -5.1 ± 0.7 |
| Hopper | Random Reward | **2021.1 ± 888.6** | 1599.6 ± 814.5 | 107.7 ± 65.5 | 178.8 ± 114.0 |
| | Random Dynamics | **2116.2 ± 618.6** | 1552.7 ± 532.8 | 5.9 ± 1.2 | 3.9 ± 2.2 |
| | Adversarial Reward | **751.4 ± 72.5** | 651.0 ± 58.0 | 29.2 ± 18.7 | 111.8 ± 62.7 |
| | Adversarial Dynamics | **931.2 ± 227.8** | 717.7 ± 204.3 | 5.9 ± 1.2 | 3.9 ± 2.2 |
| Average | | **2212.8** | 1083.6 | 630.0 | 607.1 |

In summary, the experimental results validate the theoretical impact of data corruption for value-based offline RL algorithms. Our practical implementation algorithm demonstrates superior efficacy under different data corruption types, thereby highlighting its potential for real-world applications.

# 6    Conclusion

This work investigates the adversarially corrupted offline RL with general function approximation. We propose the uncertainty weight iteration and a weighted version of PEVI. Under a partial coverage condition and a well-explored dataset, our algorithm achieves a suboptimality bound, where the corruption-independent term recovers the uncorrupted bound, and the corruption-related term nearly matches the lower bound in linear models. Furthermore, our experiments demonstrate promising results showing that our practical implementation, UWMSG, significantly enhances the performance of the state-of-the-art offline RL algorithm under reward and dynamics data corruptions.

Our work suggests several potential future directions. First, it remains unsolved whether one can design robust algorithms to handle additional distribution shifts of the state-action pairs. Second, when the corruption level is unknown, how to design a theoretically robust offline RL algorithm and overcome the lack of adaptivity in the offline setting requires further research. Finally, we hope that our work sheds light on applying uncertainty weights to improve robustness in deep offline RL works and even practical applications.

# Acknowledgements

We thank Wei Xiong for valuable discussions and feedback on an early draft of this work. We also thank the anonymous reviewers for their helpful comments.

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
