# A Proof of Theorem 1

## A.1 Step I: Suboptimality Decomposition

**Lemma A.1** (Regret Decomposition). *Assuming that $\mathcal{T}^h f_n \in \widehat{\mathcal{F}}^h$ for all $h \in [H]$, we have*

$$\text{SubOpt}(\hat{\pi}, x) \leq 2 \sum_{h=1}^{H} \beta^h \mathbb{E}_{\pi_*} \big[ b^h(x^h, a^h) \,\big|\, x^1 = x \big].$$

*Proof.* By invoking Lemma G.1, we can decompose the suboptimality as follows:

$$\begin{aligned}
\text{SubOpt}(\hat{\pi}, x) &= V_*^1(x) - V_{\hat{\pi}}^1(x) \\
&= \sum_{h=1}^{H} \mathbb{E}_{\pi_*} \big[ f_n^h(x^h, \pi_*(x^h)) - f_n^h(x^h, \hat{\pi}(x^h)) \,\big|\, x^1 = x \big] \\
&\quad - \sum_{h=1}^{H} \mathbb{E}_{\pi_*} \big[ \mathcal{E}^h(f_n, x^h, a^h) \,\big|\, x^1 = x \big] + \sum_{h=1}^{H} \mathbb{E}_{\hat{\pi}} \big[ \mathcal{E}^h(f_n, x^h, a^h) \,\big|\, x^1 = x \big] \\
&\leq \underbrace{- \sum_{h=1}^{H} \mathbb{E}_{\pi_*} \big[ \mathcal{E}^h(f_n, x^h, a^h) \,\big|\, x^1 = x \big]}_{(a)} + \underbrace{\sum_{h=1}^{H} \mathbb{E}_{\hat{\pi}} \big[ \mathcal{E}^h(f_n, x^h, a^h) \,\big|\, x^1 = x \big]}_{(b)}, \quad (14)
\end{aligned}$$

where the inequality is due to $\hat{\pi}(x^h) = \text{argmax}_{a \in \mathcal{A}} f_n^h(x^h, a)$. Then, we handle the above two terms respectively. We first tackle term (b). Supposing that $\mathcal{T}^h f_n^{h+1} \in \widehat{\mathcal{F}}^h$, we get from the definition of the confidence set in Algorithm 2 that

$$\left( \lambda + \sum_{i=1}^{n} \frac{((\mathcal{T}^h f_n^{h+1})(x_i^h, a_i^h) - \hat{f}^h(x_i^h, a_i^h))^2}{(\sigma_i^h)^2} \right)^{1/2} \leq \beta^h. \quad (15)$$

Therefore, for any $(x^h, a^h) \in \mathcal{X} \times \mathcal{A}$, we have

$$\begin{aligned}
&\big| \hat{f}^h(x^h, a^h) - (\mathcal{T}^h f_n^{h+1})(x^h, a^h) \big| \\
&= \left( \lambda + \sum_{i=1}^{n} \frac{((\mathcal{T}^h f_n^{h+1})(x_i^h, a_i^h) - \hat{f}^h(x_i^h, a_i^h))^2}{(\sigma_i^h)^2} \right)^{1/2} \cdot \frac{\big| \hat{f}^h(x^h, a^h) - (\mathcal{T}^h f_n^{h+1})(x^h, a^h) \big|}{\left( \lambda + \sum_{i=1}^{n} ((\mathcal{T}^h f_n^{h+1})(x_i^h, a_i^h) - \hat{f}^h(x_i^h, a_i^h))^2/(\sigma_i^h)^2 \right)^{1/2}} \\
&\leq \beta^h b^h(x^h, a^h),
\end{aligned}$$

where the inequality uses (15) and the definition of the bonus $b^h$ (8). Then, combining the above result with $f_n^h(x, a) = \max(0, \hat{f}^h(x, a) - \beta^h b^h(x, a))$ and $(\mathcal{T}^h f_n^{h+1})(x^h, a^h) \geq 0$, we get

$$-2\beta^h b^h(x^h, a^h) \leq f_n^h(x^h, a^h) - (\mathcal{T}^h f_n^{h+1})(x^h, a^h) \leq 0.$$

Hence, the term (b) is bounded by

$$\sum_{h=1}^{H} \mathbb{E}_{\hat{\pi}} \big[ \mathcal{E}^h(f_n, x^h, a^h) \,\big|\, x^1 = x \big] \leq 0. \quad (16)$$

Moreover, the term (a) is bounded by

$$-\sum_{h=1}^{H} \mathbb{E}_{\pi_*} \big[ \mathcal{E}^h(f_n, x^h, a^h) \,\big|\, x^1 = x \big] \leq 2 \sum_{h=1}^{H} \beta^h \mathbb{E}_{\pi_*} \big[ b^h(x^h, a^h) \,\big|\, x^1 = x \big]. \quad (17)$$

Finally, by taking (17) and (16) back into (14), we conclude the proof. $\qquad\square$

## A.2 Step II: Sharper confidence radius for Pessimism

**Lemma A.2** (Confidence Radius). *In Algorithm 2, for all $h \in [H]$ we have $\mathcal{T}^h f_n^{h+1} \in \widehat{\mathcal{F}}^h$ with probability at least $1 - \delta$, where we choose*

$$\beta^h = 24\alpha\zeta^h + \left(12\lambda + 12\ln(2HN_n^h(\gamma)/\delta) + 12(5\beta^{h+1}\gamma)^2 n + 60\beta^{h+1}\gamma\sqrt{nC_1^h(n,\zeta)}\right)^{1/2},$$

*where*

$$N_n(\gamma) = \max_h N\left(\frac{\gamma}{n}, \mathcal{F}^h\right) \cdot N\left(\frac{\gamma}{n}, \mathcal{F}^{h+1}\right) \cdot N\left(\frac{\gamma}{n}, \mathcal{B}^{h+1}(\lambda)\right),$$

*we use the notation $C_1^h(n,\zeta) = 2(\sum_{i=1}^n (\zeta_i^h)^2 + 2n\eta^2 + 3\eta^2\ln(2/\delta))$, and $\zeta^h = \sum_{i=1}^n \zeta_i^h$.*

*Proof.* At each step $h \in [H]$, let $\mathcal{F}_\gamma^{h+1}$ be a $(\gamma, \|\cdot\|_\infty)$ cover of $\mathcal{F}^{h+1}$, and $\mathcal{B}_\gamma^{h+1}$ as an $(\gamma, \|\cdot\|_\infty)$ cover of $\mathcal{B}^{h+1}(\lambda)$. Then, we construct $\bar{\mathcal{F}}_\gamma^{h+1} = \mathcal{F}_\gamma^{h+1} \oplus \beta_\tau^{h+1} \mathcal{B}_\gamma^{h+1}$ as a $((1 + \beta^{h+1})\gamma, \|\cdot\|_\infty)$ cover of $f_n^{h+1}(\cdot)$. Given $f_n^{h+1}$, we have $\bar{f}_n^{h+1} \in \bar{\mathcal{F}}_\gamma^{h+1}$ so that $\|\bar{f}^{h+1} - f_n^{h+1}\|_\infty \le \bar{\epsilon} = (1 + \beta^{h+1})\gamma$. Then, we define $y_i^h = r_i^h + f_n^{h+1}(x_i^{h+1})$, $\bar{y}_i^h = r_i^h + \bar{f}^{h+1}(x_i^{h+1})$ and

$$\tilde{f}^h = \operatorname*{argmin}_{f^h \in \mathcal{F}^h} \sum_{i=1}^n (f^h(x_i^h, a_i^h) - \bar{y}_i^h)^2.$$

We know from the definition of the covers that

$$\left(\sum_{i=1}^n (\hat{f}^h(x_i^h, a_i^h) - \bar{y}_i^h)^2\right)^{1/2} \le \left(\sum_{i=1}^n (\hat{f}^h(x_i^h, a_i^h) - y_i^h)^2\right)^{1/2} + \sqrt{n}\bar{\epsilon}$$

$$\le \left(\sum_{i=1}^n (\tilde{f}^h(x_i^h, a_i^h) - y_i^h)^2\right)^{1/2} + \sqrt{n}\bar{\epsilon} \le \left(\sum_{i=1}^n (\tilde{f}^h(x_i^h, a_i^h) - \bar{y}_i^h)^2\right)^{1/2} + 2\sqrt{n}\bar{\epsilon}, \quad (18)$$

where the first and third inequality is due to $\|\bar{f}^{h+1} - f_n^{h+1}\|_\infty \le \bar{\epsilon}$, and the second inequality comes from the fact that $\hat{f}^h$ is the ERM solution to the least squares problem. Then, we can invoke Lemma G.3 by taking $f_*$ as $\mathbb{E}[\bar{y}_i^h | x_i^h, a_i^h] = (\mathcal{T}_{\mathcal{D}}^h \bar{f}^{h+1})(x_i^h, a_i^h)$ and $f_b$ as $\mathcal{T}^h \bar{f}^{h+1}$. With probability at least $1 - \delta$, we obtain:

$$\sum_{i=1}^n \frac{\left(\hat{f}^h(x_i^h, a_i^h) - (\mathcal{T}^h \bar{f}^{h+1})(x_i^h, a_i^h)\right)^2}{(\sigma_i^h)^2}$$

$$\le 10\ln(2HN_n^h(\gamma)/\delta) + 5 \underbrace{\sum_{i=1}^n \frac{|\hat{f}^h(x_i^h, a_i^h) - (\mathcal{T}^h \bar{f}^{h+1})(x_i^h, a_i^h)| \cdot |\zeta_i^h|}{(\sigma_i^h)^2}}_{(a)}$$

$$+ 10(\gamma + 2\bar{\epsilon}) \cdot \left((\gamma + 2\bar{\epsilon})n + \sqrt{nC_1(n,\zeta)}\right), \quad (19)$$

where $C_1(n,\zeta) = 2(\zeta^2 + 2n + 3\ln(2/\delta))$.

According to Lemma 3.1, the term (a) can be controlled by the weight design:

$$\sum_{i=1}^n |\zeta_i^h| \cdot \frac{|\hat{f}^h(x_i^h, a_i^h) - (\mathcal{T}^h \bar{f}^{h+1})(x_i^h, a_i^h)|}{(\sigma_i^h)^2}$$

$$\le \zeta^h \sup_i \frac{|\hat{f}^h(x_i^h, a_i^h) - (\mathcal{T}^h \bar{f}^{h+1})(x_i^h, a_i^h)|}{(\sigma_i^h)^2}$$

$$\le 2\alpha\zeta^h \sqrt{\lambda + \sum_{i=1}^n \frac{\left(\hat{f}^h(x_i^h, a_i^h) - (\mathcal{T}^h \bar{f}^{h+1})(x_i^h, a_i^h)\right)^2}{(\sigma_i^h)^2}},$$

where the second inequality is obtained since $\sigma_i^h$ satisfies (6). Taking this result back into (19), we finally get for all $h \in [H]$,

$$\sum_{i=1}^{n} \frac{\left(\hat{f}^h(x_i^h, a_i^h) - (\mathcal{T}^h \bar{f}^{h+1})(x_i^h, a_i^h)\right)^2}{(\sigma_i^h)^2}$$

$$\leq 10 \ln(2HN_n^h(\gamma)/\delta) + 10\alpha\zeta^h\beta^h + 5\gamma\zeta + 10(\gamma + 2\bar{\epsilon}) \cdot ((\gamma + 2\bar{\epsilon})n + \sqrt{nC_1(t, \zeta)})$$

$$= 10 \ln(2HN_n^h(\gamma)/\delta) + 10\alpha\zeta^h\beta^h + 5\gamma\zeta + 10(2\beta^{h+1} + 3)^2\gamma^2 n + 10(2\beta^{h+1} + 3)\gamma\sqrt{nC_1(n, \zeta)},$$

where the last equality uses $\bar{\epsilon} = (1 + \beta^{h+1})\gamma$. Therefore, it follows that with probability at least $1 - \delta$,

$$\left(\sum_{i=1}^{n} \frac{\left(\hat{f}_n^h(x_i^h, a_i^h) - (\mathcal{T}^h f_n^{h+1})(x_i^h, a_i^h)\right)^2}{(\sigma_i^h)^2} + \lambda\right)^{1/2}$$

$$\leq \left(\sum_{i=1}^{n} \frac{\left(\hat{f}_n^h(x_i^h, a_i^h) - (\mathcal{T}^h \bar{f}_n^{h+1})(x_i^h, a_i^h)\right)^2}{(\sigma_i^h)^2}\right)^{1/2} + \sqrt{n}\bar{\epsilon} + \sqrt{\lambda}$$

$$\leq \left(10 \ln(2HN_n^h(\gamma)/\delta) + 10\alpha\zeta^h\beta^h + 5\gamma\zeta + 10(2\beta^{h+1} + 3)^2\gamma^2 n \right.$$

$$\left. + 10(2\beta^{h+1} + 3)\gamma\sqrt{nC_1(n, \zeta)}\right)^{1/2} + (\beta^{h+1} + 1)\gamma\sqrt{n} + \sqrt{\lambda}$$

$$\leq \left(12\lambda + 12 \ln(2HN_n^h(\gamma)/\delta) + 24\alpha\zeta^h\beta^h + 12(5\beta^{h+1}\gamma)^2 n + 60\beta^{h+1}\gamma\sqrt{nC_1(n, \zeta)}\right)^{1/2} \leq \beta^h.$$

Therefore, we complete the proof. □

### A.3 Step III: Bound the Suboptimality

*Proof of Theorem 1.* We know from Lemma A.1 and Lemma A.2 that with probability at least $1 - \delta$,

$$\text{SubOpt}(\hat{\pi}, x) \leq 2 \sum_{h=1}^{H} \beta^h \mathbb{E}_{\pi_*}\left[b^h(x^h, a^h) \,\middle|\, x^1 = x\right]$$

$$= 2 \sum_{h=1}^{H} \beta^h \mathbb{E}_{\pi_*}\left[\frac{b^h(x^h, a^h)}{\sigma^h(x^h, a^h)} \cdot \mathbb{1}(\sigma^h(x^h, a^h) = 1)\right.$$

$$\left. + \frac{b^h(x^h, a^h)}{\sigma^h(x^h, a^h)} \cdot \sigma^h(x^h, a^h) \cdot \mathbb{1}(\sigma^h(x^h, a^h) > 1) \,\middle|\, x^1 = x\right]$$

$$\leq 2 \sum_{h=1}^{H} \beta^h \mathbb{E}_{\pi_*}\left[\frac{b^h(x^h, a^h)}{\sigma^h(x^h, a^h)} + \left(\frac{b^h(x^h, a^h)}{\sigma^h(x^h, a^h)}\right)^2 \cdot \frac{1}{\alpha} \,\middle|\, x^1 = x\right], \qquad (20)$$

where the last inequality is deduced since from the definition of $\sigma^h(x^h, a^h)$ in (13), we get for $\sigma^h(x^h, a^h) > 1$ that

$$\sigma^h(x^h, a^h) = \frac{1}{\sigma^h(x^h, a^h)} \cdot \sup_{f, f' \in \widehat{\mathcal{F}}^h} \frac{|f(x^h, a^h) - f'(x^h, a^h)|/\alpha}{\sqrt{\lambda + \sum_{i=1}^{n}(f(x_i^h, a_i^h) - f'(x_i^h, a_i^h))^2/(\sigma_i^h)^2}} = \frac{b^h(x^h, a^h)}{\alpha\sigma^h(x^h, a^h)}.$$

From the definition of the weighted coverage condition (12), we have

$$\mathbb{E}_{\pi_*}\left[\left(\frac{b^h(x^h, a^h)}{\sigma^h(x^h, a^h)}\right)^2 \,\middle|\, x^1 = x\right] = \frac{\text{CC}^\sigma(\lambda, \widehat{\mathcal{F}}, \mathcal{Z}_n^H)}{n}.$$

Combining the above equation and $\mathbb{E}X \leq \sqrt{\mathbb{E}X^2}$, we can bound (20) by

$$\sum_{h=1}^{H} \beta^h \mathbb{E}_{\pi_*}\left[\frac{b^h(x^h,a^h)}{\sigma^h(x^h,a^h)} + \left(\frac{b^h(x^h,a^h)}{\sigma^h(x^h,a^h)}\right)^2 \cdot \frac{1}{\alpha}\,\Big|\, x^1 = x\right]$$

$$\leq \sum_{h=1}^{H} \beta^h \left[\sqrt{\frac{\mathrm{CC}^\sigma(\lambda,\widehat{\mathcal{F}},\mathcal{Z}_n^H)}{n}} + \frac{\mathrm{CC}^\sigma(\lambda,\widehat{\mathcal{F}},\mathcal{Z}_n^H)}{n} \cdot \frac{1}{\alpha}\right]$$

$$\leq \sqrt{\frac{\mathrm{CC}^\sigma(\lambda,\widehat{\mathcal{F}},\mathcal{Z}_n^H)}{n}} \sum_{h=1}^{H} \beta^h + \frac{\mathrm{CC}^\sigma(\lambda,\widehat{\mathcal{F}},\mathcal{Z}_n^H)}{n} \cdot \frac{\sum_{h=1}^{H} \beta^h}{\alpha}$$

$$\leq \sqrt{\frac{\mathrm{CC}^\sigma(\lambda,\widehat{\mathcal{F}},\mathcal{Z}_n^H)}{n}} \cdot c_\beta\left(\alpha\zeta + H\sqrt{\ln N}\right) + \frac{\mathrm{CC}^\sigma(\lambda,\widehat{\mathcal{F}},\mathcal{Z}_n^H)}{n} \cdot c_\beta\left(\zeta + \frac{H\sqrt{\ln N}}{\alpha}\right).$$

By choosing $\alpha = H\sqrt{\ln N}/\zeta$, we can obtain the result:

$$\mathrm{SubOpt}(\hat{\pi},x) = \tilde{\mathcal{O}}\left(\frac{3H\sqrt{\mathrm{CC}^\sigma(\lambda,\widehat{\mathcal{F}},\mathcal{Z}_n^H)\cdot\ln N}}{\sqrt{n}} + \frac{\zeta\cdot\mathrm{CC}^\sigma(\lambda,\widehat{\mathcal{F}},\mathcal{Z}_n^H)}{n}\right).$$

Ultimately, we can invoke Lemma 4.1 to obtain that with probability at least $1 - 2\delta$,

$$\mathrm{SubOpt}(\hat{\pi},x) = \tilde{\mathcal{O}}\left(\frac{H(\mathrm{CC}(\lambda,\widehat{\mathcal{F}},\mathcal{Z}_n^H))^{1/4}\cdot(\ln N_n(\gamma))^{1/2}}{n^{1/2}(C(\widehat{\mathcal{F}},\mu))^{1/4}} + \frac{\zeta(\mathrm{CC}(\lambda,\widehat{\mathcal{F}},\mathcal{Z}_n^H))^{1/2}}{n(C(\widehat{\mathcal{F}},\mu))^{1/2}}\right),$$

which concludes the proof. $\qquad\square$

# B  Proofs of Auxiliary Results

## B.1  Proof of Lemma 3.1

To begin with, we demonstrate that the uncertainty weight iteration (Algorithm 1) converges, thus satisfying the approximate condition in (6).

*Proof of Lemma 3.1.* We will demonstrate this result via the convergence of monotone real number sequences. To begin with, we prove the monotonicity of $\{(\sigma_i^t)^2\}_{t=0}^\infty$ by induction. When $\tau = 0$, we know that $(\sigma_i^1)^2 \geq 1 = \sigma_i^0$ for $i \in [n]$. When $\tau = t$, assume that $(\sigma_i^t)^2 \geq (\sigma_i^{t-1})^2$ for all $i \in [n]$, which implies that

$$\sup_{f,f'\in\mathcal{F}} \frac{|f(x_i,a_i) - f'(x_i,a_i)|/(\alpha)}{\sqrt{\lambda + \sum_{j=1}^n (f(x_j,a_j) - f'(x_j,a_j))^2/(\sigma_j^t)^2}}$$

$$\geq \sup_{f,f'\in\mathcal{F}} \frac{|f(x_i,a_i) - f'(x_i,a_i)|/(\alpha)}{\sqrt{\lambda + \sum_{j=1}^n (f(x_j,a_j) - f'(x_j,a_j))^2/(\sigma_j^{t-1})^2}}.$$

Therefore, we get $(\sigma_i^{t+1})^2 \geq (\sigma_i^t)^2$ for all $i \in [n]$. Then, when $t = T$, we deduce from $(\sigma_i^{T+1})^2 \geq (\sigma_i^T)^2$ that for each $i \in [n]$,

$$(\sigma_i^T)^2 \leq (\sigma_i^{T+1})^2 \leq \max\left(1, \sup_{f,f'\in\mathcal{F}} \frac{|f(x_i,a_i) - f'(x_i,a_i)|/\alpha}{\sqrt{\lambda + \sum_{j=1}^n (f(x_j,a_j) - f'(x_j,a_j))^2/(\sigma_j^N)^2}}\right),$$

which implies the second inequality of (6).

Then, we obtain the upper bounds of each sequence: for $i \in [n]$ and any $t \geq 0$

$$\sup_{f,f'\in\mathcal{F}} \frac{|f(x_i,a_i) - f'(x_i,a_i)|/\alpha}{\sqrt{\lambda + \sum_{j=1}^n (f(x_j,a_j) - f'(x_j,a_j))^2/(\sigma_j^t)^2}} \leq \frac{1}{\alpha\sqrt{\lambda}},$$

where we use $f(\cdot) \in [0,1]$ for any $f \in \mathcal{F}$. Thus, each $\{(\sigma_i^t)^2\}_{t=0}^{\infty}$ has a $\max(1, 1/(\alpha\sqrt{\lambda}))$ upper bound.

According to the convergence of monotone real number sequences, the sequence $\{(\sigma_i^t)^2\}_{t=1}^{\infty}$ converges for all $i \in [n]$, which implies that $\{\log((\sigma_i^t)^2)\}_{t=1}^{\infty}$ converges. We know from the definition of convergence that for any $\mu > 0$, there exists an $N(\mu)$ such that for any $t \geq N$,

$$\log\left((\sigma_i^{t+1})^2\right) - \log\left((\sigma_i^t)^2\right) \leq \log(1+\mu),$$

which implies that

$$\frac{(\sigma_i^{t+1})^2}{(\sigma_i^n)^2} \leq 1 + \mu. \tag{21}$$

For any $t \geq N$, if $\sigma_i^{t+1} = 1$, we have from the monotonicity that $\sigma_i^t = 1$, thus satisfying the first inequality of (6). If $\sigma_i^{n+1} > 1$, we deduce from (21) that

$$(\sigma_i^t)^2 \geq \frac{1}{1+\mu}(\sigma_i^{t+1})^2 = \frac{1}{1+\mu} \sup_{f,f' \in \mathcal{F}} \frac{|f(x_i, a_i) - f'(x_i, a_i)|/(\alpha)}{\sqrt{\lambda + \sum_{j=1}^{n}(f(x_j, a_j) - f'(x_j, a_j))^2/(\sigma_j^t)^2}}.$$

Hence, the inequality is proved by taking $\mu = 1$ and stop the iteration at round $t = N+1$. $\qquad\square$

## B.2 Connection between Coverage Coefficients

In this part, we state that in the linear MDP, the coverage condition in Jin et al. [24] implies our coverage assumption. Recall the coverage coefficient defined in (9):

$$\mathrm{CC}(\lambda, \widehat{\mathcal{F}}, \mathcal{Z}_n^H) = \max_{h \in [H]} \mathbb{E}_{\pi_*}\left[ \sup_{f,f' \in \widehat{\mathcal{F}}^h} \frac{n(f(x^h, a^h) - f'(x^h, a^h))^2}{\lambda + \sum_{i=1}^{n}(f(x_i^h, a_i^h) - f'(x_i^h, a_i^h))^2} \,\bigg|\, x^1 = x \right].$$

When the function space $\mathcal{F}^h$ is embedded into a $d$-dimensional vector space: $\mathcal{F}^h = \{\langle w(f), \phi(\cdot)\rangle : z \to \mathbb{R}\}$, where $z$ denotes the state-action pair $(x, a)$. Then, we define $\Lambda^h = \lambda I + \sum_{i=1}^{n} \phi(z_i^h)\phi(z_i^h)^\top$.

The coverage condition in Jin et al. [24] assumes that there exists a constant $c^\dagger$ such that for all $h \in [H]$,

$$\Lambda^h \succeq I + c^\dagger n \mathbb{E}_{\pi_*}\left[\phi(z^h)\phi(z^h)^\top \,\big|\, x^1 = x\right]. \tag{22}$$

**Lemma B.1.** *In the linear setting with dimension $d$, if the coverage condition in (22) holds with a finite constant $c^\dagger > 0$, the coverage coefficient in (9) is also finite:*

$$\mathrm{CC}(\lambda, \widehat{\mathcal{F}}, \mathcal{Z}_n^H) \leq \frac{d}{c^\dagger} < \infty.$$

*Proof.* We deduce that

$$\sup_{f,f' \in \widehat{\mathcal{F}}^h} \frac{n\big(\langle w(f) - w(f'), \phi(z)\rangle\big)^2}{\lambda + \sum_{i=1}^{n}\big(\langle w(f) - w(f'), \phi(z_i^h)\rangle\big)^2}$$

$$\leq \sup_{f,f' \in \widehat{\mathcal{F}}^h} \frac{n\big(\langle w(f) - w(f'), \phi(z)\rangle\big)^2}{\big(w(f) - w(f')\big)^\top \Lambda^h \big(w(f) - w(f')\big)}$$

$$\leq n\phi(z)^\top (\Lambda^h)^{-1}\phi(z),$$

where the second inequality uses

$$\langle w(f) - w(f'), \phi(z)\rangle \leq \sqrt{\big(w(f) - w(f')\big)^\top \Lambda^h \big(w(f) - w(f')\big)} \cdot \sqrt{\phi(z)^\top (\Lambda^h)^{-1}\phi(z)}.$$

Thus, the coverage coefficient (9) for the linear model is bounded by

$$\mathrm{CC}(\lambda, \widehat{\mathcal{F}}, \mathcal{Z}_n^H) \leq \max_{h \in [H]} \mathbb{E}_{\pi_*}\left[ \sup_{f,f' \in \widehat{\mathcal{F}}^h} \frac{n\big(\langle w(f) - w(f'), \phi(z)\rangle\big)^2}{\lambda + \sum_{i=1}^{n}\big(\langle w(f) - w(f'), \phi(z_i^h)\rangle\big)^2} \,\bigg|\, x^1 = x \right]$$

$$\leq \max_{h \in [H]} \mathbb{E}_{\pi_*}\left[n\phi(z^h)^\top (\Lambda^h)^{-1}\phi(z^h) \,\big|\, x^1 = x\right].$$

If the sufficient "coverage" in (22) holds, then, we have

$$\mathrm{CC}(\lambda, \widehat{\mathcal{F}}, \mathcal{Z}_n^H) \leq \frac{1}{c^\dagger} \max_{h \in [H]} \mathrm{Tr}\left\{ (\Lambda^h)^{-1} \cdot \left( I + c^\dagger n \mathbb{E}_{\pi_*}\left[ \phi(z^h)\phi(z^h)^\top \mid x^1 = x \right] \right) \right\}$$

$$\leq \frac{d}{c^\dagger} < \infty,$$

which concludes the proof. $\qquad\square$

## B.3   Connections between Weighted and Unweighted Coverage Coefficients

We use the shorthand notation $z = (x, a)$ for any $(x, a) \in \mathcal{X} \times \mathcal{A}$. Recall the definition of weighted coverage coefficient in (12):

$$\mathrm{CC}^\sigma(\lambda, \widehat{\mathcal{F}}, \mathcal{Z}_n^H) = \max_{h \in [H]} \mathbb{E}_{\pi_*}\left[ \sup_{f, f' \in \widehat{\mathcal{F}}^h} \frac{n(f(z^h) - f'(z^h))^2 / (\sigma^h(z^h))^2}{\lambda + \sum_{i=1}^n (f(z_i^h) - f'(z_i^h))^2 / (\sigma_i^h)^2} \,\middle|\, x^1 = x \right],$$

where

$$(\sigma^h(z^h))^2 = \max\left( 1, \sup_{f, f' \in \widehat{\mathcal{F}}^h} \frac{|f(z^h) - f'(z^h)| / \alpha}{\sqrt{\lambda + \sum_{i=1}^n (f(z_i^h) - f'(z_i^h))^2 / (\sigma_i^h)^2}} \right).$$

In the sequel, we explore the relationship between the weighted coverage coefficient $\mathrm{CC}^\sigma(\lambda, \widehat{\mathcal{F}}, \mathcal{Z}_n^H)$ and the unweighted coverage coefficient $\mathrm{CC}(\lambda, \widehat{\mathcal{F}}, \mathcal{Z}_n^H)$ defined in (9):

$$\mathrm{CC}(\lambda, \widehat{\mathcal{F}}, \mathcal{Z}_n^H) = \max_{h \in [H]} \mathbb{E}_{\pi_*}\left[ \sup_{f, f' \in \widehat{\mathcal{F}}^h} \frac{n(f(z^h) - f'(z^h))^2}{\lambda + \sum_{i=1}^n (f(z_i^h) - f'(z_i^h))^2} \,\middle|\, x^1 = x \right].$$

Let $\mu^h$ be the empirical data distribution:

$$\sum_{i=1}^n (f(z_i^h) - f'(z_i^h))^2 / (\sigma_i^h)^2 = n \sum_{z^h \in \mathcal{X} \times \mathcal{A}} \mu(z^h) \cdot \frac{(f(z^h) - f'(z^h))^2}{(\sigma^h(z^h))^2}.$$

Now, we present the proof of Lemma 4.1.

*Proof of Lemma 4.1.* According to the proof of Lemma 3.1, for each $i \in [n], h \in [H]$, the weight $(\sigma_i^h)^2$ is upper bounded by

$$(\sigma_i^h)^2 \leq \max(1, 1/(\alpha\sqrt{\lambda})).$$

If $\alpha\sqrt{\lambda} > 1$, we have

$$\mathrm{CC}^\sigma(\lambda, \widehat{\mathcal{F}}, \mathcal{Z}_n^H) = \mathrm{CC}(\lambda, \widehat{\mathcal{F}}, \mathcal{Z}_n^H).$$

If $\alpha\sqrt{\lambda} \leq 1$, we have

$$\mathrm{CC}^\sigma(\lambda, \widehat{\mathcal{F}}, \mathcal{Z}_n^H) = \max_{h \in [H]} \sum_{z^h} d_{\pi_*}^h(z^h) \sup_{f, f' \in \widehat{\mathcal{F}}^h} \frac{n(f(z^h) - f'(z^h))^2 / (\sigma^h(z^h))^2}{\lambda + n \sum_{\bar{z}^h} \mu(\bar{z}^h)(f(\bar{z}^h) - f'(\bar{z}^h))^2 / (\sigma^h(\bar{z}^h))^2}$$

$$\leq \max_{h \in [H]} \sum_{z^h} d_{\pi_*}^h(z^h) \sup_{f, f' \in \widehat{\mathcal{F}}^h} \frac{(f(z^h) - f'(z^h))^2 / (\sigma^h(z^h))^2}{\lambda/n + \alpha\sqrt{\lambda} \sum_{\bar{z}^h} \mu(\bar{z}^h)(f(\bar{z}^h) - f'(\bar{z}^h))^2}$$

$$\leq \frac{1}{\alpha\sqrt{\lambda}} \max_{h \in [H]} \sum_{z^h} d_{\pi_*}^h(z^h) \sup_{f, f' \in \widehat{\mathcal{F}}^h} \frac{(f(z^h) - f'(z^h))^2 / (\sigma^h(z^h))^2}{\lambda/n + \sum_{\bar{z}^h} \mu(\bar{z}^h)(f(\bar{z}^h) - f'(\bar{z}^h))^2}, \quad (23)$$

where we omit the condition $\mid x^1 = x$ from the distribution $d_{\pi_*}^h(\cdot \mid x^1 = x)$ when there is no confusion, and the last inequality uses $\alpha\sqrt{\lambda} \leq 1$. For any $z^h \sim d_{\pi_*}^h(z^h)$, we take the $f_{z^h}, f'_{z^h} \in \widehat{\mathcal{F}}^h$ that maximize the term:

$$\frac{(f_{z^h}(z^h) - f'_{z^h}(z^h))^2 / (\sigma^h(z^h))^2}{\lambda/n + \sum_{\bar{z}^h} \mu(\bar{z}^h)(f_{z^h}(\bar{z}^h) - f'_{z^h}(\bar{z}^h))^2}. \quad (24)$$

Then, we can write (23) as

$$\mathrm{CC}^\sigma(\lambda, \widehat{\mathcal{F}}, \mathcal{Z}_n^H) \le \frac{1}{\alpha\sqrt{\lambda}} \max_{h\in[H]} \sum_{z^h} d_{\pi_*}^h(z^h) \frac{(f_{z^h}(z^h) - f'_{z^h}(z^h))^2/(\sigma^h(z^h))^2}{\lambda/n + \sum_{\bar{z}^h} \mu(\bar{z}^h)(f_{z^h}(\bar{z}^h) - f'_{z^h}(\bar{z}^h))^2}. \qquad (25)$$

To further bound (25), we need to lower bound $(\sigma^h(z^h))^2$ in (13) by

$$\sup_{f,f'\in\widehat{\mathcal{F}}^h} \frac{|f(z^h) - f'(z^h)|/\alpha}{\sqrt{\lambda + \sum_{i=1}^n (f(z_i^h) - f'(z_i^h))^2/(\sigma_i^h)^2}} \ge \frac{|f_{z^h}(z^h) - f'_{z^h}(z^h)|/\alpha}{\sqrt{\lambda + \sum_{i=1}^n (f_{z^h}(z_i^h) - f'_{z^h}(z_i^h))^2/(\sigma_i^h)^2}}$$

$$\ge \frac{|f_{z^h}(z^h) - f'_{z^h}(z^h)|}{2\alpha\beta^h},$$

where the last inequality is due to the confidence interval:

$$\lambda + \sum_{i=1}^n \frac{(f_{z^h}(z_i^h) - f'_{z^h}(z_i^h))^2}{(\sigma_i^h)^2}$$

$$= \lambda + \sum_{i=1}^n \frac{(f_{z^h}(z_i^h) - \hat{f}(z_i^h) + \hat{f}(z_i^h) - f'_{z^h}(z_i^h))^2}{(\sigma_i^h)^2}$$

$$\le 2\Big[\lambda + \sum_{i=1}^n \frac{(f_{z^h}(z_i^h) - \hat{f}(z_i^h))^2}{(\sigma_i^h)^2} + \lambda + \sum_{i=1}^n \frac{\hat{f}(z_i^h) - f'_{z^h}(z_i^h))^2}{(\sigma_i^h)^2}\Big] \le 4(\beta^h)^2.$$

Hence, by taking this lower bound into (25) and taking $\beta^h = C_\beta\sqrt{\ln N} = C_\beta\sqrt{\lambda}$, we get

$$\mathrm{CC}^\sigma(\lambda, \widehat{\mathcal{F}}, \mathcal{Z}_n^H)$$

$$\le \frac{2\alpha\beta^h}{\alpha\sqrt{\lambda}} \max_{h\in[H]} \sum_{z^h} d_{\pi_*}^h(z^h) \frac{|f_{z^h}(z^h) - f'_{z^h}(z^h)|}{\lambda/n + \sum_{\bar{z}^h} \mu^h(\bar{z}^h)(f_{z^h}(\bar{z}^h) - f'_{z^h}(\bar{z}^h))^2}$$

$$= 2C_\beta \max_{h\in[H]} \underbrace{\sum_{z^h} d_{\pi_*}^h(z^h) \frac{|f_{z^h}(z^h) - f'_{z^h}(z^h)|}{\lambda/n + \sum_{\bar{z}^h} \mu^h(\bar{z}^h)(f_{z^h}(\bar{z}^h) - f'_{z^h}(\bar{z}^h))^2} \cdot \mathbb{1}\big(\|f_{z^h} - f'_{z^h}\|_\infty \le n^{-1}\big)}_{(a)}$$

$$+ 2C_\beta \max_{h\in[H]} \sum_{z^h} d_{\pi_*}^h(z^h) \frac{|f_{z^h}(z^h) - f'_{z^h}(z^h)|}{\lambda/n + \sum_{\bar{z}^h} \mu^h(\bar{z}^h)(f_{z^h}(\bar{z}^h) - f'_{z^h}(\bar{z}^h))^2} \cdot \mathbb{1}\big(\|f_{z^h} - f'_{z^h}\|_\infty > n^{-1}\big).$$

$$(26)$$

The term (a) can be bounded by

$$(a) \le \sum_{z^h} d_{\pi_*}^h(z^h) \frac{n^{-1}}{\lambda n^{-1}} = \frac{1}{\lambda},$$

which is taken back into (26) to obtain

$$\mathrm{CC}^{\sigma}(\lambda, \widehat{\mathcal{F}}, \mathcal{Z}_n^H)$$

$$\leq \frac{2C_\beta}{\lambda} + 2C_\beta \max_{h\in[H]} \sum_{z^h} d^h_{\pi_*}(z^h) \frac{|f_{z^h}(z^h) - f'_{z^h}(z^h)|}{\lambda/n + \sum_{\bar{z}^h} \mu^h(\bar{z}^h)(f_{z^h}(\bar{z}^h) - f'_{z^h}(\bar{z}^h))^2} \cdot \mathbb{1}\big(\|f_{z^h} - f'_{z^h}\|_\infty > n^{-1}\big)$$

$$\leq \frac{2C_\beta}{\lambda} + 2C_\beta \max_{h\in[H]} \sum_{z^h} \frac{\sqrt{d^h_{\pi_*}(z^h)}}{\sqrt{\lambda/n + \sum_{\bar{z}^h} \mu^h(\bar{z}^h)(f_{z^h}(\bar{z}^h) - f'_{z^h}(\bar{z}^h))^2}}$$

$$\cdot \frac{\sqrt{d^h_{\pi_*}(z^h)}|f_{z^h}(z^h) - f'_{z^h}(z^h)|}{\sqrt{\lambda/n + \sum_{\bar{z}^h} \mu^h(\bar{z}^h)(f_{z^h}(\bar{z}^h) - f'_{z^h}(\bar{z}^h))^2}} \cdot \mathbb{1}\big(\|f_{z^h} - f'_{z^h}\|_\infty > n^{-1}\big)$$

$$\leq \frac{2C_\beta}{\lambda} + 2C_\beta \max_{h\in[H]} \underbrace{\bigg(\sum_{z^h} \frac{d^h_{\pi_*}(z^h)}{\lambda/n + \sum_{\bar{z}^h} \mu^h(\bar{z}^h)(f_{z^h}(\bar{z}^h) - f'_{z^h}(\bar{z}^h))^2} \cdot \mathbb{1}\big(\|f_{z^h} - f'_{z^h}\|_\infty > n^{-1}\big)\bigg)^{1/2}}_{I_1}$$

$$\cdot \underbrace{\bigg(\max_{h\in[H]} \sum_{z^h} d^h_{\pi_*}(z^h) \frac{(f_{z^h}(z^h) - f'_{z^h}(z^h))^2}{\lambda/n + \sum_{\bar{z}^h} \mu^h(\bar{z}^h)(f_{z^h}(\bar{z}^h) - f'_{z^h}(\bar{z}^h))^2}\bigg)^{1/2}}_{I_2}, \tag{27}$$

where the second equality is by the Cauchy-Schwarz inequality. Then, we handle the terms $I_1$ and $I_2$ separately. For the term $I_2$, since $f_{z^h}, f'_{z^h}$ maximize the term (24), they also maximize the term

$$\frac{(f(z^h) - f'(z^h))^2}{\lambda/n + \sum_{\bar{z}^h} \mu^h(\bar{z}^h)(f(\bar{z}^h) - f'(\bar{z}^h))^2},$$

which indicates that $I_2$ can be written as

$$I_2 = \max_{h\in[H]} \sum_{z^h} d^h_{\pi_*}(z^h) \sup_{f,f'\in\widehat{\mathcal{F}}^h} \frac{n(f(z^h) - f'(z^h))^2}{\lambda + \sum_{i=1}^n (f(z_i^h) - f'(z_i^h))^2}$$

$$\leq \mathrm{CC}(\lambda, \widehat{\mathcal{F}}, \mathcal{Z}_n^H).$$

To bound the term $I_1$, for any distinct $f, f' \in \widehat{\mathcal{F}}^h$ and $\|f - f'\|_\infty > n^{-1}$, there exists two distinct functions $\bar{f}, \bar{f}' \in \bar{\mathcal{F}}^h$ such that $\|f - \bar{f}\|_\infty \leq n^{-1}$ and $\|f' - \bar{f}'\|_\infty \leq n^{-1}$. Then, we get

$$\bigg|\sum_{z^h} \mu^h(z)\big(f(z^h) - f'(z^h)\big)^2 - \sum_{z^h} \mu^h(z)\big(\bar{f}(z^h) - \bar{f}'(z^h)\big)^2\bigg|$$

$$\leq \bigg|\sum_{z^h} \mu^h(z)\big(f(z^h) - \bar{f}(z^h) - (f'(z^h) - \bar{f}'(z^h))\big)\big(f(z^h) - f'(z^h) + \bar{f}(z^h) - \bar{f}'(z^h)\big)\bigg|$$

$$\leq \sum_{z^h} \mu^h(z^h) \cdot \frac{2}{n} \cdot 2 = \frac{4}{n},$$

where the last inequality is because $\bar{f}, \bar{f}'$ satisfy (10) and $f \in [0, 1]$ for any $f \in \mathcal{F}^h$. It follows that

$$\sum_{z^h} \mu^h(z)\big(f(z^h) - f'(z^h)\big)^2 \geq \sum_{z^h} \mu^h(z)\big(\bar{f}(z^h) - \bar{f}'(z^h)\big)^2 - \frac{4}{n}$$

$$\geq C(\widehat{\mathcal{F}}, \mu) - \frac{4}{n} \geq \frac{C(\widehat{\mathcal{F}}, \mu)}{2}. \tag{28}$$

Therefore, we apply the above result with $f = f_{z^h}$, $f' = f'_{z^h}$ to bound the term $I_1$ as

$$I_1 \leq \sum_{z^h} \frac{d^h_{\pi_*}(z^h)}{C(\widehat{\mathcal{F}}, \mu)/2} = \frac{2}{C(\widehat{\mathcal{F}}, \mu)},$$

where the last equality uses $\sum_{z^h} d^h_{\pi_*}(z^h) = 1$. Therefore, by taking the bound of terms $I_1, I_2$ into (27), we have

$$\mathrm{CC}^\sigma(\lambda, \widehat{\mathcal{F}}, \mathcal{Z}^H_n) = \frac{2C_\beta}{\lambda} + 2C_\beta \sqrt{\frac{2\mathrm{CC}(\lambda, \widehat{\mathcal{F}}, \mathcal{Z}^H_n)}{C(\widehat{\mathcal{F}}, \mu)}} = \tilde{\mathcal{O}}\Big(\sqrt{\frac{\mathrm{CC}(\lambda, \widehat{\mathcal{F}}, \mathcal{Z}^H_n)}{C(\widehat{\mathcal{F}}, \mu)}}\Big).$$

Finally, we will analyze the relationship between $\mathrm{CC}(\lambda, \widehat{\mathcal{F}}, \mathcal{Z}^H_n)$ and $C(\widehat{\mathcal{F}}, \mu)$. Let $f_{z^h}, f'_{z^h} \in \widehat{\mathcal{F}}^h$ be the maximizer of

$$\frac{n(f(z^h) - f'(z^h))^2}{\lambda + \sum_{i=1}^n (f(z^h_i) - f'(z^h_i))^2}.$$

Then, we have

$$\mathrm{CC}(\lambda, \widehat{\mathcal{F}}, \mathcal{Z}^H_n)$$

$$= \max_{h \in [H]} \sum_{z^h} d^h_{\pi_*}(z^h) \frac{(f_{z^h}(z^h) - f'_{z^h}(z^h))^2}{\lambda/n + \sum_{\bar{z}^h} \mu^h(\bar{z}^h)(f_{z^h}(\bar{z}^h) - f'_{z^h}(\bar{z}^h))^2} \cdot \mathbb{1}\big(\|f_{z^h} - f'_{z^h}\|_\infty \le n^{-1}\big)$$

$$+ \max_{h \in [H]} \sum_{z^h} d^h_{\pi_*}(z^h) \frac{(f_{z^h}(z^h) - f'_{z^h}(z^h))^2}{\lambda/n + \sum_{\bar{z}^h} \mu^h(\bar{z}^h)(f_{z^h}(\bar{z}^h) - f'_{z^h}(\bar{z}^h))^2} \cdot \mathbb{1}\big(\|f_{z^h} - f'_{z^h}\|_\infty > n^{-1}\big)$$

$$\le \frac{1}{\lambda n} + \sum_{z^h} \frac{d^h_{\pi_*}(z^h)}{C(\widehat{\mathcal{F}}, \mu)/2}$$

$$= O((C(\widehat{\mathcal{F}}, \mu))^{-1}),$$

where the inequality holds due to (28). It follows that

$$\mathrm{CC}(\lambda, \widehat{\mathcal{F}}, \mathcal{Z}^H_n) = O\Big(\sqrt{\mathrm{CC}(\lambda, \widehat{\mathcal{F}}, \mathcal{Z}^H_n)/C(\widehat{\mathcal{F}}, \mu)}\Big),$$

which concludes the proof. $\qquad\square$

**Interpretation of Assumption 4.1 in linear MDPs.** Now, we illustrate the condition (10) with the linear model. When the function space $\mathcal{F}^h$ can be embedded into a $d$-dimensional vector space: $\mathcal{F}^h = \{\langle w(f), \phi(\cdot)\rangle : z \to \mathbb{R}\}$, the condition (10) becomes: for any two distinct $f, f' \in \bar{\mathcal{F}}^h$,

$$\sum_{z^h} \mu^h(z)\big(f(z^h) - f'(z^h)\big)^2 = \big(w(f) - w(f')\big)^\top \bar{\Lambda}^h \big(w(f) - w(f')\big) \ge C(\widehat{\mathcal{F}}, \mu). \quad (29)$$

where $\mu^h$ is the data empirical distribution.

In the following lemma, we demonstrate that the above condition holds as long as the learner has excess to a well-explored dataset (30), which is a wildly-adopted assumption in the literature of offline linear MDPs [11, 46, 63]. Note that $d^{-1}$ is the largest possible order of the minimum eigenvalue since for any data distribution $\tilde{\mu}^h$, $\sigma_{\min}(\mathbb{E}_{z^h \sim \mu^h}[\phi(z)\phi(z)^\top]) \le d^{-1}$ by using $\|\phi(z)\| \le 1$ for any $z \in \mathcal{X} \times \mathcal{A}$.

**Lemma B.2.** *In the linear setting, if we assume that the data distributions $\mu^h$ satisfy the following minimum eigenvalue condition: there exists an absolute constant $\bar{c} > 0$ such that*

$$\sigma_{\min}\Big(\mathbb{E}_{z^h \sim \mu^h}[\phi(z)\phi(z)^\top]\Big) = \frac{\bar{c}}{d}, \quad (30)$$

*the dataset $\mathcal{D}$ consists of $n \ge 128d^2\bar{c}^{-2}\log(d/(2\delta))$ independent trajectories, then, the condition (29) with $C(\widehat{\mathcal{F}}, \mu) = a\bar{c}/(2d)$ will holds with probability at least $1 - \delta$.*

*Proof.* To begin with, we aim to prove that the empirical matrix $\bar{\Lambda}^h = n^{-1}\sum_{i=1}^n \phi(z^h_i)\phi(z^h_i)^\top$ is positive definite with high probability. Since $\|\phi(z)\| \le 1$ for any $z \in \mathcal{X} \times \mathcal{A}$, we have for each $i \in [n]$,

$$\big(\phi(z^h_i)\phi(z^h_i)^\top - \mathbb{E}_{z^h \sim \mu^h}[\phi(z)\phi(z)^\top]\big)^2 \preceq (2I)^2.$$

By invoking the matrix Hoeffding's concentration in Lemma G.4 with $X_i = \phi(z_i^h)\phi(z_i^h)^\top - \mathbb{E}_{z^h \sim \mu^h}[\phi(z)\phi(z)^\top]$, $A_i = 2I$, $\sigma^2 = 4n$, we obtain

$$\mathbb{P}\left(\left\|\sum_{i=1}^n \left(\phi(z_i^h)\phi(z_i^h)^\top - \mathbb{E}_{z^h \sim \mu^h}[\phi(z)\phi(z)^\top]\right)\right\|_{\text{op}} \geq t\right) \leq 2d \cdot e^{-t^2/(32n)}.$$

For any $\delta > 0$, by taking $t = \sqrt{32n\log(d/\delta)}$, we have with probability at least $1 - \delta$,

$$\left\|\frac{1}{n}\sum_{i=1}^n \left(\phi(z_i^h)\phi(z_i^h)^\top - \mathbb{E}_{z^h \sim \mu^h}[\phi(z)\phi(z)^\top]\right)\right\|_{\text{op}} \leq \sqrt{\frac{32\log(d/(2\delta))}{n}}.$$

Hence, whenever $n \geq 128d^2\bar{c}^{-2}\log(d/(2\delta))$, by combing the above result with (30), we have with probability at least $1 - \delta$,

$$\sigma_{\min}(\bar{\Lambda}^h) = \sigma_{\min}\left(\frac{1}{n}\sum_{i=1}^n \phi(z_i^h)\phi(z_i^h)^\top\right) \geq \frac{\bar{c}}{2d}. \tag{31}$$

Then, since the cardinality of the cover $\bar{\mathcal{F}}^h$ is $\tilde{\mathcal{O}}(d)$, we can define $a = \min_{f,f' \bar{\mathcal{F}}^h}\|w(f) - w(f')\|^2$. Thus, by using (31), the condition (29) is inferred: for any $f, f'\bar{\mathcal{F}}^h$ with probability at least $1 - \delta$,

$$\left(w(f) - w(f')\right)^\top \bar{\Lambda}^h\left(w(f) - w(f')\right) \geq \frac{\bar{c}}{2d}\|w(f) - w(f')\|^2 \geq \frac{a\bar{c}}{2d} = C(\hat{\mathcal{F}}, \mu).$$

$\square$

## B.4 Lower Bound for Linear MDPs with Corruption

*Proof of Theorem 2.* For any dimension $d$, step horizon $H$ and corruption level $\zeta$, we construct a tabular MDP with action number $A > 2$ and state number $S \leq (A/2)^{H/2}$ such that $SA = d$ by following the proof of Theorem 3.1 in [60]. Particularly, since we assume that $V^h(x) \in [0, 1]$ for any $h \in [H]$ and $x \in \mathcal{X}$, we set all the rewards equaling to 1 to $1/H$. Then, by taking the corruption fraction as $\epsilon = \zeta/n$, an $\epsilon$-contamination adversary can perturb all the chosen rewards from $1/H$ to 0. Thus, the learner must suffer from at least $\Omega(SA\epsilon) = \Omega(SA\zeta n^{-1})$ suboptimality with probability at least $1/4$.

$\square$

## B.5 Relationship between the Bootstrapped Uncertainty and Bonus

In the sequel, we discuss the relationship between the bootstrapped uncertainty and the bonus function by considering linear function approximation. Using $z$ to denote the feature variable of state-action pair $(x, a)$, we estimate the Q-value function $Q_\phi$ by $Q_{w^h}^h(z) = (w^h)^\top \phi(z)$ to minimize the Bellman error target with weights:

$$\hat{w}^h = \operatorname*{argmin}_{w \in \mathbb{R}^d} \sum_{z_i^h \in \mathcal{D}} \frac{\left((\hat{\mathcal{T}}^h Q_w)(z_i^h) - w^\top \phi(z_i^h)\right)^2}{\sigma(z_i^h)}, \tag{32}$$

where $\{\sigma(z_i^h) > 0\}$ is a group of predetermined weights and for any $z_i^h \in \mathcal{D}$,

$$(\hat{\mathcal{T}}^h Q_w)(z_i^h) = r^h(z_i^h) + V_w^{h+1}(z_i^{h+1}).$$

Additionally, we define the noise in this weighted least square problem as $\epsilon = \hat{\mathcal{T}}^h Q_w(x, a) - Q_w^h(z)$.

**Bonus functions.** In the traditional linear MDP, we often use the following term as the bonus function:

$$b_L^h(z) = \sqrt{\phi(z)^\top (\Lambda^h)^{-1}\phi(z)}, \tag{33}$$

where $\Lambda^h = \sum_{i=1}^n \phi(z_i^h)\phi(z_i^h)^\top/(\sigma_i^h)^2$, and we use the shorthand notation for any matrix $A$ and vector $x$: $\|x\|_A = \sqrt{x^\top Ax}$.

The bonus function in the general form (8) turns into the following form under the linear setting:

$$b^h(z) = \sup_{f,f' \in \widehat{\mathcal{F}}^h} \frac{\left| \left(w(f) - w(f')\right)^\top \phi(z) \right|}{\sqrt{\lambda + \sum_{i=1}^n \left( (w(f) - w(f'))^\top \phi(z_i^h)/\sigma_i^h \right)^2}}.$$

We demonstrate that the linear and general forms of bonus functions are almost equivalent under mild conditions.

**Lemma B.3.** *Under the linear MDP, if the function space is broad enough such that for any $z \in \mathcal{X} \times \mathcal{A}$, there exists $f, f' \in \widehat{\mathcal{F}}^h$ satisfying that $w(f) - w(f')$ and $(\Lambda^h)^{-1}\phi(z)$ are in the same direction and not too close, i.e., for some $\alpha > 0$,*

$$w(f) - w(f') = \alpha \cdot (\Lambda^h)^{-1}\phi(z), \quad and \quad \|w(f) - w(f')\|^2_{\Lambda^h} \geq \lambda,$$

*then, we have for any $z \in \mathcal{X} \times \mathcal{A}$,*

$$\frac{b_L^h(z)}{\lambda^{1/4} + 1} \leq b^h(z) \leq b_L^h(z).$$

*Proof.* First, we will prove $b^h(z) \leq b_L^h(z)$. By the definition of $b^h$, we have

$$\begin{aligned}
b^h(z) &= \sup_{f,f' \in \widehat{\mathcal{F}}^h} \frac{\left| \left(w(f) - w(f')\right)^\top \phi(z) \right|}{\sqrt{\lambda + \sum_{i=1}^n \left( (w(f) - w(f'))^\top (\phi(z_i^h)/\sigma_i^h)^2 \right)}} \\
&\leq \sup_{f,f' \in \widehat{\mathcal{F}}^h} \frac{\left| \left(w(f) - w(f')\right)^\top \phi(z) \right|}{\sqrt{\left(w(f) - w(f')\right)^\top \Lambda^h \left(w(f) - w(f')\right)}} \\
&\leq \sqrt{\phi(z)^\top (\Lambda^h)^{-1}\phi(z)} = b_L^h(z).
\end{aligned}$$

Then, we will prove $b^h(z) \geq b_L^h(z)$. By the assumption, for any $z \in \mathcal{X} \times \mathcal{A}$, there exists $f_1, f_2 \in \widehat{\mathcal{F}}^h$ such that for some $\alpha > 0$,

$$w(f_1) - w(f_2) = \alpha \cdot (\Lambda^h)^{-1}\phi(z),$$

which implies that

$$(\Lambda^h)^{1/2}(w(f_1) - w(f_2)) = \alpha \cdot (\Lambda^h)^{-1/2}\phi(z).$$

Then, we have

$$\begin{aligned}
b^h(z) &= \sup_{f,f' \in \widehat{\mathcal{F}}^h} \frac{\left| \left(w(f) - w(f')\right)^\top \phi(z) \right|}{\sqrt{\lambda + \left(w(f) - w(f')\right)^\top \Lambda^h \left(w(f) - w(f')\right)}} \\
&\geq \frac{\left| \left(w(f_1) - w(f_2)\right)^\top \phi(z) \right|}{\sqrt{\lambda + \left(w(f_1) - w(f_2)\right)^\top \Lambda^h \left(w(f_1) - w(f_2)\right)}} \\
&= \frac{\left\| (\Lambda^h)^{1/2}(w(f_1) - w(f_2)) \right\| \cdot \left\| (\Lambda^h)^{-1/2}\phi(z) \right\|}{\|w(f) - w(f')\|_{\Lambda^h}} \cdot \frac{\|w(f) - w(f')\|_{\Lambda^h}}{\sqrt{\lambda + \|w(f) - w(f')\|^2_{\Lambda^h}}} \\
&= \frac{\left\| (\Lambda^h)^{1/2}(w(f_1) - w(f_2)) \right\| \cdot \left\| (\Lambda^h)^{-1/2}\phi(z) \right\|}{\|w(f) - w(f')\|_{\Lambda^h}} \cdot \frac{1}{\sqrt{\lambda/\|w(f) - w(f')\|^2_{\Lambda^h} + 1}}. \quad (34)
\end{aligned}$$

Since $\|w(f) - w(f')\|^2_{\Lambda^h} \geq \sqrt{\lambda}$, we have

$$\sqrt{\frac{\lambda}{\|w(f) - w(f')\|^2_{\Lambda^h}} + 1} \leq \sqrt{\lambda^{1/2} + 1} \leq \lambda^{1/4} + 1.$$

Taking this result back into (34) leads to

$$b^h(a) \geq \left\| (\Lambda^h)^{-1/2} \phi(z) \right\| \cdot \frac{1}{\sqrt{\lambda / \|w(f) - w(f')\|_{\Lambda^h}^2 + 1}}$$

$$\geq \frac{\sqrt{\phi(z)^\top (\Lambda^h)^{-1} \phi(z)}}{\lambda^{1/4} + 1} = \frac{b_L^h(z)}{\lambda^{1/4} + 1}.$$

Therefore, we conclude the proof. $\qquad\square$

**Connection between the bootstrapped uncertainty and bonus functions.** We begin with illustrating the equivalence between the bootstrapped uncertainty and the linear form of the bonus in the following lemma. Since we actually compute the uncertainty weights for only single iteration, we let $\sigma_i^h = 1$.

**Lemma B.4.** *For any $z \in \mathcal{X} \times \mathcal{A}$,*

$$Var_{\hat{w}^h}(Q_{\hat{w}^h}^h(z)) = Var_{\hat{w}^h}(z^\top \hat{w}^h) = z^\top (\Lambda^h)^{-1} z.$$

*Proof.* Let $y_i^h = r^h(z_i^h) + V_w^{h+1}(z_i^{h+1})$. Under the assumption that $\epsilon_i^h \sim N(0,1)$, since the closed form solution to the problem is

$$\hat{w}^h = (\Lambda^h)^{-1} \sum_{z_i^h \in \mathcal{D}} y_i^h z_i^h = (\Lambda^h)^{-1} \sum_{z_i^h \in \mathcal{D}} (Q_\phi^h(z_i^h) + \epsilon_i^h) z_i^h,$$

we obtain that

$$\hat{w}^h | \mathcal{D} \sim N(\mu^h, (\Lambda^h)^{-1}),$$

where

$$\mu^h = (\Lambda^h)^{-1} \sum_{z_i^h \in \mathcal{D}} Q_\phi^h(z_i^h) z_i^h, \quad \Lambda^h = \sum_{i=1}^n z_i^h (z_i^h)^\top.$$

Then, it follows that for any $z \in \mathcal{A} \times \mathcal{A}$,

$$\mathrm{Var}_{\hat{w}^h}(Q_{\hat{w}^h}^h(z) | \mathcal{D}) = \mathrm{Var}_{\hat{w}^h}(z^\top \hat{w}^h | \mathcal{D}) = z^\top (\Lambda^h)^{-1} z.$$

Hence, we complete the proof. $\qquad\square$

In Lemma B.4, we find that the standard deviation of the $Q$-value function is equivalent to the linear LCB-bonus $b_L^h(z) = \sqrt{z^\top (\Lambda^h)^{-1} z}$. Moreover, recall that the bootstrapped uncertainty $\mathbb{V}_{j=1,\dots,N}[Q_{w_j}]$ is the standard deviation of the bootstrapped $Q$-value functions. Therefore, according to Osband et al. [34] our proposed bootstrapped uncertainty can serve as an estimation for the bonus function $b_L^h$ under the linear MDP setting. Theoretically, we can use the uncertainty weight iteration (Algorithm 1) to construct the weighted bootstrap uncertainty.

By combining Lemma B.3 and Lemma B.4, we conclude that by taking a sufficiently broad function approximation class $\widehat{\mathcal{F}}^h$ and sufficiently small parameter $\lambda$, the proposed bootstrapped uncertainty is an estimation of the general form of the bonus in linear MDPs. More importantly, in the experiments, the estimation of the uncertainty is simplified and shares the spirit with the theoretical analysis due to two reasons: 1) due to the complexity of the nonlinear version of uncertainty, which is expressed as

$$\sup_{f, f' \in \mathcal{F}} \frac{|f(z_i) - f'(z_i)|}{\sqrt{\lambda + \sum_{j=1}^n (f(z_j) - f'(z_j))^2 / \sigma_j^2}},$$

we simplify the estimation by using the bootstrap uncertainty, which is an unbiased estimation of uncertainty in the linear version, and 2) combining the uncertainty weight iteration and the bootstrap uncertainty estimation is cumbersome, so we only iterate once during simulations.

# C  Results for Distribution Shift

In this section, we consider an MDP$(\mathcal{X}, \mathcal{A}, H, \mathbb{P}, r)$ and an offline dataset $\mathcal{D} = \{(x_i^h, a_i^h)\}_{i,h=1}^{H}$ with adversarial corruption and distribution shift. Specifically, for each trajectory $\{(x_i^h, a_i^h)\}_{h=1}^{H}$, we define $\rho_i > 0$ to measure the ditribution shift of this trajectory. For example, when the learner's goal varies from the training data, the goal shift can be embedded into the initial state $x^1$ and captured by $\rho(x^1)$. Hence, we define a new notion of corruption level $\zeta$, capturing both adversarial corruption and distribution shift.

**Definition C.1** (Cumulative Corruption). *The cumulative corruption is $\zeta$ if at any step $h \in [H]$, for any sequence $\{x_i^h, a_i^h\}_{i,h=1}^{n,H} \subset \mathcal{X} \times \mathcal{A}$, $\{\rho_i\}_{i=1}^{n} \subset \mathbb{R}^+$ and function $\{g^h : \mathcal{X} \to [0,1]\}_{h=1}^{H}$, we have*

$$\zeta = \sum_{h=1}^{H} \sum_{i=1}^{n} \rho_i |\zeta_i^h|, \quad \zeta_i^h = (\mathcal{T}^h g - \mathcal{T}_{\mathcal{D}}^h g)(x_i^h, a_i^h).$$

---

**Algorithm 3** Uncertainty Weight Iteration with Distribution Shift

1: **Input:** $\{(x_i, a_i, \rho_i)\}_{i=1}^{n}, \mathcal{F}, \alpha > 0$
2: **Initialization:** $t = 0$, $\sigma_i^0 = 1$, $i = 1, \ldots, n$
3: **repeat**
4:    $t \leftarrow t + 1$
5:    $(\sigma_i^t)^2 \leftarrow \max \left( 1, \sup_{f,f' \in \mathcal{F}} \frac{|f(x_i,a_i) - f'(x_i,a_i)|/(\alpha\rho_i)}{\sqrt{\lambda + \sum_{j=1}^{n}(f(x_j,a_j) - f'(x_j,a_j))^2/(\sigma_j^{t-1})^2}} \right)$, $i = 1, \ldots, n$
6: **until** $\max_{i \in [n]} \left( \sigma_i^t / \sigma_i^{t-1} \right)^2 \leq 2$
7: **Output:** $\{\sigma_i^t\}_{i=1}^{n}$

---

The main challenge in handling distribution shifts is the new weight design. We propose uncertainty weight iteration with distribution shift in Algorithm 3, where we put $\rho_i$ on the denominator. Similarly, we can follow Lemma 3.1 to demonstrate that the iteration converges and the output weights $\{\sigma_i := \sigma_i^N\}_{i=1}^{n}$ satisfy

$$\sigma_i^2 \geq \max \left( 1, \frac{1}{2} \sup_{f,f' \in \mathcal{F}} \frac{|f(x_i,a_i) - f'(x_i,a_i)|/(\alpha\rho_i)}{\sqrt{\lambda + \sum_{j=1}^{n}(f(x_j,a_j) - f'(x_j,a_j))^2/\sigma_j^2}} \right). \tag{35}$$

Therefore, just by replacing the weight iteration (Algorithm 1) in CR-PEVI Algorithm 2 with Algorithm 3, we obtain an algorithm robust to both corruption and distribution shift, named as **CORDS-PEVI**. Because CORDS-PEVI highly repeats Algorithm 2, we do not present the pseudo-code of the algorithm.

Then, the suboptimality bound achieved by CORDS-PEVI is presented in the following theorem.

**Theorem 3.** *If the coverage coefficient in Definition 4.1 is finite and Assumption 4.1 holds, under CORDS-PEVI, for any cumulative corruption $\zeta$ and $\delta > 0$, we choose the covering parameter $\gamma = 1/(n \max_h \beta^h \zeta^h)$, the eluder parameter $\lambda = \ln(N_n(\gamma))$, the weighting parameter $\alpha = H\sqrt{\ln N_n(\gamma)}/\zeta$, and the confidence radius*

$$\beta^h = c_\beta \left( \alpha\zeta^h + \sqrt{\ln(HN_n(\gamma)/\delta)} \right) \text{ for } h = H, \ldots, 1,$$

*where*

$$N_n(\gamma) = \max_h N\left(\frac{\gamma}{n}, \mathcal{F}^h\right) \cdot N\left(\frac{\gamma}{n}, \mathcal{F}^{h+1}\right) \cdot N\left(\frac{\gamma}{n}, \mathcal{B}^{h+1}(\lambda)\right).$$

*Then, with probability at least $1 - 3\delta$, the sub-optimality is bounded by*

$$\text{SubOpt}(\hat{\pi}, x) = \tilde{\mathcal{O}}\left( \frac{H(\text{CC}(\lambda, \widehat{\mathcal{F}}, \mathcal{Z}_n^H))^{1/4} \cdot (\ln N_n(\gamma))^{1/2}}{n^{1/2}(C(\widehat{\mathcal{F}}, \mu))^{1/4}} + \frac{\zeta(\text{CC}(\lambda, \widehat{\mathcal{F}}, \mathcal{Z}_n^H))^{1/2}}{n(C(\widehat{\mathcal{F}}, \mu))^{1/2}} \right).$$

## C.1  Analysis of the Result

The main difference in the analysis between the model with and without a distribution shift is the bound for the confidence radius.

### C.1.1 Sharp bound of the confidence radius.

**Lemma C.1** (Confidence Radius). *In CORDS-PEVI, for all $h \in [H]$ we have $\mathcal{T}^h f_n^{h+1} \in \widehat{\mathcal{F}}^h$ with probability at least $1 - \delta$, where*

$$\beta^h = 12\alpha\zeta^h + \left(12\lambda + 12\ln(2HN_n^h(\gamma)/\delta) + 12(5\beta^{h+1}\gamma)^2 n + 60\beta^{h+1}\gamma\sqrt{nC_1^h(n,\zeta)}\right)^{1/2},$$

*we use the notation*

$$N_n(\gamma) = \max_h N\left(\frac{\gamma}{n}, \mathcal{F}^h\right) \cdot N\left(\frac{\gamma}{n}, \mathcal{F}^{h+1}\right) \cdot N\left(\frac{\gamma}{n}, \mathcal{B}^{h+1}(\lambda)\right),$$

*and $C_1^h(n,\zeta) = 2(\sum_{i=1}^n (\zeta_i^h)^2 + 2n\eta^2 + 3\eta^2 \ln(2/\delta))$, $\zeta^h = \sum_{i=1}^n \rho_i \zeta_i^h$.*

*Proof.* We use similar methods as the proof of Lemma A.2. At each step $h \in [H]$, by notating $\mathcal{F}_\gamma^{h+1}$ as a $(\gamma, \|\cdot\|_\infty)$ cover of $\mathcal{F}^{h+1}$, and $\mathcal{B}_\gamma^{h+1}$ as a $(\gamma, \|\cdot\|_\infty)$ cover of $\mathcal{B}^{h+1}(\lambda)$, we construct $\bar{\mathcal{F}}_\gamma^{h+1} = \mathcal{F}_\gamma^{h+1} \oplus \beta_\tau^{h+1}\mathcal{B}_\gamma^{h+1}$ as a $((1+\beta^{h+1})\gamma, \|\cdot\|_\infty)$ cover of $f_n^{h+1}(\cdot)$. For the $f_n^{h+1}$, there exists a $\bar{f}^{h+1} \in \bar{\mathcal{F}}_\gamma^{h+1}$ such that $\|\bar{f}^{h+1} - f_n^{h+1}\|_\infty \leq \bar{\epsilon} = (1+\beta^{h+1})\gamma$. Then, we define $y_i^h = r_i^h + f_n^{h+1}(x_i^{h+1})$, $\bar{y}_i^h = r_i^h + \bar{f}^{h+1}(x_i^{h+1})$ and

$$\tilde{f}^h = \operatorname*{argmin}_{f^h \in \mathcal{F}^h} \sum_{i=1}^n (f^h(x_i^h, a_i^h) - \bar{y}_i^h)^2.$$

Since (18) also holds true, we can invoke Lemma G.3 by taking $f_*$ as $\mathbb{E}[\bar{y}_i^h | x_i^h, a_i^h] = (\mathcal{T}_\mathcal{D}^h \bar{f}^{h+1})(x_i^h, a_i^h)$ and $f_b$ as $\mathcal{T}^h \bar{f}^{h+1}$. With probability at least $1 - \delta$, we obtain:

$$\sum_{i=1}^n \frac{\left(\hat{f}^h(x_i^h, a_i^h) - (\mathcal{T}^h \bar{f}^{h+1})(x_i^h, a_i^h)\right)^2}{(\sigma_i^h)^2}$$

$$\leq 10\ln(2HN_n^h(\gamma)/\delta) + 5\underbrace{\sum_{i=1}^n \frac{|\hat{f}^h(x_i^h, a_i^h) - (\mathcal{T}^h \bar{f}^{h+1})(x_i^h, a_i^h)| \cdot |\zeta_i^h|}{(\sigma_i^h)^2}}_{(a)}$$

$$+ 10(\gamma + 2\bar{\epsilon}) \cdot \left((\gamma + 2\bar{\epsilon})n + \sqrt{nC_1(n,\zeta)}\right), \tag{36}$$

where $C_1(n,\zeta) = 2(\zeta^2 + 2n + 3\ln(2/\delta))$.

From the weight design and Lemma 3.1, term (a) is bounded by

$$\sum_{i=1}^n |\zeta_i^h| \cdot \frac{|\hat{f}^h(x_i^h, a_i^h) - (\mathcal{T}^h \bar{f}^{h+1})(x_i^h, a_i^h)|}{(\sigma_i^h)^2} \leq \sum_{i=1}^n \rho_i |\zeta_i^h| \cdot \frac{|\hat{f}^h(x_i^h, a_i^h) - (\mathcal{T}^h \bar{f}^{h+1})(x_i^h, a_i^h)|}{\rho_i(\sigma_i^h)^2}$$

$$\leq \zeta^h \sup_i \frac{|\hat{f}^h(x_i^h, a_i^h) - (\mathcal{T}^h \bar{f}^{h+1})(x_i^h, a_i^h)|}{\rho_i(\sigma_i^h)^2}$$

$$\leq 2\alpha\zeta^h \sqrt{\lambda + \sum_{i=1}^n \frac{\left(\hat{f}^h(x_i^h, a_i^h) - (\mathcal{T}^h \bar{f}^{h+1})(x_i^h, a_i^h)\right)^2}{(\sigma_i^h)^2}}$$

where the last inequality is obtained since $\sigma_i^h$ satisfies (35). Taking this result back into (36), we finally get for all $h \in [H]$,

$$\sum_{i=1}^n \frac{\left(\hat{f}^h(x_i^h, a_i^h) - (\mathcal{T}^h \bar{f}^{h+1})(x_i^h, a_i^h)\right)^2}{(\sigma_i^h)^2}$$

$$\leq 10\ln(2HN_n^h(\gamma)/\delta) + 10\alpha\zeta^h\beta^h + 5\gamma\zeta + 10(\gamma + 2\bar{\epsilon}) \cdot ((\gamma + 2\bar{\epsilon})n + \sqrt{nC_1(t,\zeta)})$$

$$= 10\ln(2HN_n^h(\gamma)/\delta) + 10\alpha\zeta^h\beta^h + 5\gamma\zeta + 10(2\beta^{h+1} + 3)^2\gamma^2 n + 10(2\beta^{h+1} + 3)\gamma\sqrt{nC_1(n,\zeta)},$$

where the last euqlaity uses $\bar{\epsilon} = (1 + \beta^{h+1})\gamma$. Therefore, it follows that with probability at least $1 - \delta$,

$$\left( \sum_{i=1}^{n} \frac{\left( \hat{f}_n^h(x_i^h, a_i^h) - (\mathcal{T}^h f_n^{h+1})(x_i^h, a_i^h) \right)^2}{(\sigma_i^h)^2} + \lambda \right)^{1/2}$$

$$\leq \left( \sum_{i=1}^{n} \frac{\left( \hat{f}_n^h(x_i^h, a_i^h) - (\mathcal{T}^h \bar{f}_n^{h+1})(x_i^h, a_i^h) \right)^2}{(\sigma_i^h)^2} \right)^{1/2} + \sqrt{n}\bar{\epsilon} + \sqrt{\lambda}$$

$$\leq \left( 10\ln(2HN_n^h(\gamma)/\delta) + 10\alpha\zeta^h\beta^h + 5\gamma\zeta + 10(2\beta^{h+1} + 3)^2\gamma^2 n \right.$$

$$\left. + 10(2\beta^{h+1} + 3)\gamma\sqrt{nC_1(n, \zeta)} \right)^{1/2} + (\beta^{h+1} + 1)\gamma\sqrt{n} + \sqrt{\lambda}$$

$$\leq \left( 12\lambda + 12\ln(2HN_n^h(\gamma)/\delta) + 24\alpha\zeta^h\beta^h + 12(5\beta^{h+1}\gamma)^2 n + 60\beta^{h+1}\gamma\sqrt{nC_1(n, \zeta)} \right)^{1/2}$$

$$\leq \beta^h,$$

which finishes the proof. □

### C.1.2 Connections between weighted and unweighted coefficient.

We define the coverage coefficient incorporating the uncertainty weights under distribution shift as

$$\mathrm{CC}^\sigma(\lambda, \widehat{\mathcal{F}}, \mathcal{Z}_n^H) = \max_{h \in [H]} \mathbb{E}_{\pi_*}\left[ \sup_{f, f' \in \widehat{\mathcal{F}}^h} \frac{n(f(x^h, a^h) - f'(x^h, a^h))^2/\sigma^h(x^h, a^h)^2}{\lambda + \sum_{i=1}^{n}(f(x_i^h, a_i^h) - f'(x_i^h, a_i^h))^2/(\sigma_i^h)^2} \,\bigg|\, x^1 = x \right], \tag{37}$$

where the weight for the trajectory induced by the optimal policy is

$$(\sigma_*^h(x^h, a^h))^2 = \max\left( 1, \sup_{f, f' \in \widehat{\mathcal{F}}^h} \frac{|f(x^h, a^h) - f'(x^h, a^h)|/\alpha}{\sqrt{\lambda + \sum_{i=1}^{n}(f(x_i^h, a_i^h) - f'(x_i^h, a_i^h))^2/(\sigma_i^h)^2}} \right), \tag{38}$$

where we do not consider the distribution shift for the expected trajectory induced by the optimal policy $\pi_*$.

**Lemma C.2.** Let $\rho_{\min} = \min_{i \in [n]} \rho_i$. Under CORDS-PEVI, Assumption 4.1, and the $\beta^h = C_\beta\sqrt{\ln N}$ (where $C_\beta > 0$ contains the logarithmic terms that can be omitted) and $\lambda$ given in Theorem 3, we have

$$\mathrm{CC}^\sigma(\lambda, \widehat{\mathcal{F}}, \mathcal{Z}_n^H) = \tilde{\mathcal{O}}\left((\mathrm{CC}(\lambda, \widehat{\mathcal{F}}, \mathcal{Z}_n^H))^{1/2} \cdot (C(\widehat{\mathcal{F}}, \mu))^{-1/2}\right).$$

*Proof.* We adopt the same approaches in the proof of Lemma 4.1. For each $i \in [n], h \in [H]$, since the weight $(\sigma_i^h)^2$ yielded by Algorithm 3 is upper bounded by $1/(\alpha\sqrt{\lambda}\rho_{\min})$, we get

$$\mathrm{CC}^\sigma(\lambda, \widehat{\mathcal{F}}, \mathcal{Z}_n^H) = \max_{h \in [H]} \sum_{z^h} d_{\pi_*}^h(z^h) \sup_{f, f' \in \widehat{\mathcal{F}}^h} \frac{n(f(z^h) - f'(z^h))^2/(\sigma_*^h(z^h))^2}{\lambda + n\sum_{\bar{z}^h} \mu(z^h)(f(\bar{z}^h) - f'(\bar{z}^h))^2/(\sigma^h(\bar{z}^h))^2}$$

$$\leq \max_{h \in [H]} \sum_{z^h} d_{\pi_*}^h(z^h) \sup_{f, f' \in \widehat{\mathcal{F}}^h} \frac{(f(z^h) - f'(z^h))^2/(\sigma_*^h(z^h))^2}{\lambda/n + \alpha\sqrt{\lambda}\sum_{\bar{z}^h} \mu(\bar{z}^h)(f(\bar{z}^h) - f'(\bar{z}^h))^2}$$

$$\leq \frac{1}{\alpha\rho_{\min}\sqrt{\lambda}} \max_{h \in [H]} \sum_{z^h} d_{\pi_*}^h(z^h) \sup_{f, f' \in \widehat{\mathcal{F}}^h} \frac{(f(z^h) - f'(z^h))^2/(\sigma_*^h(z^h))^2}{\lambda/n + \sum_{\bar{z}^h} \mu(\bar{z}^h)(f(\bar{z}^h) - f'(\bar{z}^h))^2}, \tag{39}$$

where the last inequality uses $\alpha\rho_{\min}\sqrt{\lambda} \leq 1$. For any $z^h \sim d_{\pi_*}^h(z^h)$, take the $f_{z^h}, f'_{z^h} \in \widehat{\mathcal{F}}^h$ that maximize the term:

$$\frac{(f_{z^h}(z^h) - f'_{z^h}(z^h))^2/(\sigma_*^h(z^h))^2}{\lambda/n + \sum_{\bar{z}^h} \mu(\bar{z}^h)(f_{z^h}(\bar{z}^h) - f'_{z^h}(\bar{z}^h))^2}. \tag{40}$$

Then, (39) is written as

$$\mathrm{CC}^\sigma(\lambda, \widehat{\mathcal{F}}, \mathcal{Z}_n^H) \leq \frac{1}{\alpha \rho_{\min} \sqrt{\lambda}} \max_{h \in [H]} \sum_{z^h} d_{\pi_*}^h(z^h) \frac{(f_{z^h}(z^h) - f'_{z^h}(z^h))^2/(\sigma_*^h(z^h))^2}{\lambda/n + \sum_{\bar{z}^h} \mu(\bar{z}^h)(f_{z^h}(\bar{z}^h) - f'_{z^h}(\bar{z}^h))^2}$$

$$= \frac{1}{\alpha \rho_{\min} \sqrt{\lambda}} \max_{h \in [H]} \sum_{z^h} d_{\pi_*}^h(z^h) \frac{(f_{z^h}(z^h) - f'_{z^h}(z^h))^2/(\sigma_*^h(z^h))^2}{\lambda/n + \sum_{\bar{z}^h} \mu(\bar{z}^h)(f_{z^h}(\bar{z}^h) - f'_{z^h}(\bar{z}^h))^2}. \quad (41)$$

To bound the above term, we need to lower bound $(\sigma_*^h(z^h))^2$ in (38) by

$$\sup_{f,f' \in \widehat{\mathcal{F}}^h} \frac{|f(z^h) - f'(z^h)|/\alpha}{\sqrt{\lambda + \sum_{i=1}^n (f(z_i^h) - f'(z_i^h))^2/(\sigma_i^h)^2}} \geq \frac{|f_{z^h}(z^h) - f'_{z^h}(z^h)|/\alpha}{\sqrt{\lambda + \sum_{i=1}^n (f_{z^h}(z_i^h) - f'_{z^h}(z_i^h))^2/(\sigma_i^h)^2}}$$

$$\geq \frac{|f_{z^h}(z^h) - f'_{z^h}(z^h)|}{2\alpha\beta^h},$$

where the last inequality uses $f_{z^h}, f_{z^h} \in \widehat{\mathcal{F}}^h$:

$$\lambda + \sum_{i=1}^n \frac{(f_{z^h}(z_i^h) - f'_{z^h}(z_i^h))^2}{(\sigma_i^h)^2} = \lambda + \sum_{i=1}^n \frac{(f_{z^h}(z_i^h) - \hat{f}(z_i^h) + \hat{f}(z_i^h) - f'_{z^h}(z_i^h))^2}{(\sigma_i^h)^2}$$

$$\leq 2\Big[\lambda + \sum_{i=1}^n \frac{(f_{z^h}(z_i^h) - \hat{f}(z_i^h))^2}{(\sigma_i^h)^2} + \lambda + \sum_{i=1}^n \frac{\hat{f}(z_i^h) - f'_{z^h}(z_i^h))^2}{(\sigma_i^h)^2}\Big] \leq 4(\beta^h)^2.$$

Thus, substituting this lower bound into (41) and taking $\beta^h = C_\beta \sqrt{\ln N} = C_\beta \sqrt{\lambda}$, we get

$$\mathrm{CC}^\sigma(\lambda, \widehat{\mathcal{F}}, \mathcal{Z}_n^H) \leq \frac{2\alpha\beta^h}{\alpha\rho_{\min}\sqrt{\lambda}} \max_{h \in [H]} \sum_{z^h} d_{\pi_*}^h(z^h) \frac{|f_{z^h}(z^h) - f'_{z^h}(z^h)|}{\lambda/n + \sum_{\bar{z}^h} \mu^h(\bar{z}^h)(f_{z^h}(\bar{z}^h) - f'_{z^h}(\bar{z}^h))^2}$$

$$= \frac{2C_\beta}{\rho_{\min}} \max_{h \in [H]} \sum_{z^h} d_{\pi_*}^h(z^h) \frac{|f_{z^h}(z^h) - f'_{z^h}(z^h)| \cdot \mathbb{1}\big(\|f_{z^h} - f'_{z^h}\|_\infty \leq n^{-1}\big)}{\lambda/n + \sum_{\bar{z}^h} \mu^h(\bar{z}^h)(f_{z^h}(\bar{z}^h) - f'_{z^h}(\bar{z}^h))^2}$$

$$+ \frac{2C_\beta}{\rho_{\min}} \max_{h \in [H]} \sum_{z^h} d_{\pi_*}^h(z^h) \frac{|f_{z^h}(z^h) - f'_{z^h}(z^h)| \cdot \mathbb{1}\big(\|f_{z^h} - f'_{z^h}\|_\infty > n^{-1}\big)}{\lambda/n + \sum_{\bar{z}^h} \mu^h(\bar{z}^h)(f_{z^h}(\bar{z}^h) - f'_{z^h}(\bar{z}^h))^2}$$

$$\leq \frac{2C_\beta}{\lambda\rho_{\min}} + \frac{2C_\beta}{\rho_{\min}} \max_{h \in [H]} \sum_{z^h} d_{\pi_*}^h(z^h) \frac{|f_{z^h}(z^h) - f'_{z^h}(z^h)| \cdot \mathbb{1}\big(\|f_{z^h} - f'_{z^h}\|_\infty > n^{-1}\big)}{\lambda/n + \sum_{\bar{z}^h} \mu^h(\bar{z}^h)(f_{z^h}(\bar{z}^h) - f'_{z^h}(\bar{z}^h))^2}.$$

Then, we can invoke the Cauchy-Schwarz inequality to split the above term into two parts:

$$\mathrm{CC}^\sigma(\lambda, \widehat{\mathcal{F}}, \mathcal{Z}_n^H) \leq \frac{2C_\beta}{\lambda\rho_{\min}} + \frac{2C_\beta}{\rho_{\min}} \max_{h \in [H]} \sum_{z^h} \frac{\sqrt{d_{\pi_*}^h(z^h)}}{\sqrt{\lambda/n + \sum_{\bar{z}^h} \mu^h(\bar{z}^h)(f_{z^h}(\bar{z}^h) - f'_{z^h}(\bar{z}^h))^2}}$$

$$\cdot \frac{\sqrt{d_{\pi_*}^h(z^h)}|f_{z^h}(z^h) - f'_{z^h}(z^h)|}{\sqrt{\lambda/n + \sum_{\bar{z}^h} \mu^h(\bar{z}^h)(f_{z^h}(\bar{z}^h) - f'_{z^h}(\bar{z}^h))^2}} \cdot \mathbb{1}\big(\|f_{z^h} - f'_{z^h}\|_\infty > n^{-1}\big)$$

$$\leq \frac{2C_\beta}{\lambda\rho_{\min}} + \frac{2C_\beta}{\rho_{\min}} \max_{h \in [H]} \underbrace{\Big(\sum_{z^h} \frac{d_{\pi_*}^h(z^h) \cdot \mathbb{1}\big(\|f_{z^h} - f'_{z^h}\|_\infty > n^{-1}\big)}{\lambda/n + \sum_{\bar{z}^h} \mu^h(\bar{z}^h)(f_{z^h}(\bar{z}^h) - f'_{z^h}(\bar{z}^h))^2}\Big)^{1/2}}_{I_1}$$

$$\cdot \underbrace{\Big(\max_{h \in [H]} \sum_{z^h} d_{\pi_*}^h(z^h) \frac{(f_{z^h}(z^h) - f'_{z^h}(z^h))^2}{\lambda/n + \sum_{\bar{z}^h} \mu^h(\bar{z}^h)(f_{z^h}(\bar{z}^h) - f'_{z^h}(\bar{z}^h))^2}\Big)^{1/2}}_{I_2}, \quad (42)$$

Then, the terms $I_1$ and $I_2$ can be handled in the same way as the proof of Lemma 4.1. For the term $I_1$, we get

$$I_1 \leq \sum_{z^h} \frac{d_{\pi_*}^h(z^h)}{C(\widehat{\mathcal{F}}, \mu)/2} = \frac{2}{C(\widehat{\mathcal{F}}, \mu)}.$$

For the term $I_2$, we have

$$I_2 = \max_{h \in [H]} \sum_{z^h} d^h_{\pi_*}(z^h) \sup_{f,f' \in \widehat{\mathcal{F}}^h} \frac{n(f(z^h) - f'(z^h))^2}{\lambda + \sum_{i=1}^n (f(z_i^h) - f'(z_i^h))^2} = \mathrm{CC}(\lambda, \widehat{\mathcal{F}}, \mathcal{Z}_n^H).$$

Therefore, by taking the bound of the terms $I_1, I_2$ into (42), we have

$$\mathrm{CC}^\sigma(\lambda, \widehat{\mathcal{F}}, \mathcal{Z}_n^H) = \frac{2C_\beta}{\lambda \rho_{\min}} + \frac{2C_\beta}{\rho_{\min}} \sqrt{\frac{2\mathrm{CC}(\lambda, \widehat{\mathcal{F}}, \mathcal{Z}_n^H)}{C(\widehat{\mathcal{F}}, \mu)}} = \tilde{\mathcal{O}}\Big(\sqrt{\frac{\mathrm{CC}(\lambda, \widehat{\mathcal{F}}, \mathcal{Z}_n^H)}{C(\widehat{\mathcal{F}}, \mu)}}\Big),$$

which concludes the proof. $\qquad\square$

### C.1.3  The suboptimality bound.

Having the guarantee for the confidence radius and the connection between weighted and unweighted coverage coefficient, we can follow the three steps in the proof of Theorem 1 to bound the suboptimality.

*Proof of Theorem 3.* Since when $\sigma_*^h(x^h, a^h) > 1$, we have

$$\sigma_*^h(x^h, a^h) = \frac{1}{\sigma_*^h(x^h, a^h)} \cdot \sup_{f,f' \in \widehat{\mathcal{F}}^h} \frac{|f(x^h, a^h) - f'(x^h, a^h)|/\alpha}{\sqrt{\lambda + \sum_{i=1}^n (f(x_i^h, a_i^h) - f'(x_i^h, a_i^h))^2/(\sigma_i^h)^2}} = \frac{b^h(x^h, a^h)}{\alpha \sigma_*^h(x^h, a^h)},$$

from which it follows that

$$\sum_{h=1}^H \beta^h \mathbb{E}_{\tilde{\pi}_*} \big[ b^h(x^h, a^h) \,\big|\, x^1 = x \big]$$

$$\leq \sum_{h=1}^H \beta^h \mathbb{E}_{\tilde{\pi}_*} \left[ \frac{b^h(x^h, a^h)}{\sigma_*^h(x^h, a^h)} + \Big( \frac{b^h(x^h, a^h)}{\sigma_*^h(x^h, a^h)} \Big)^2 \cdot \frac{1}{\alpha} \,\bigg|\, x^1 = x \right]$$

$$\leq \sum_{h=1}^H \left[ \beta^h \cdot \sqrt{\frac{\mathrm{CC}^\sigma(\lambda, \widehat{\mathcal{F}}, \mathcal{Z}_n^H)}{n}} + \frac{\beta^h}{\alpha} \cdot \frac{\mathrm{CC}^\sigma(\lambda, \widehat{\mathcal{F}}, \mathcal{Z}_n^H)}{n} \right],$$

where the last inequality uses $\mathbb{E}X \leq \sqrt{\mathbb{E}X^2}$ and

$$\mathbb{E}_{\pi_*} \left[ \Big( \frac{b^h(x^h, a^h)}{\sigma_*^h(x^h, a^h)} \Big)^2 \,\bigg|\, x^1 = x \right] = \frac{\mathrm{CC}^\sigma(\lambda, \widehat{\mathcal{F}}, \mathcal{Z}_n^H)}{n}.$$

We further get with probability at least $1 - \delta$,

$$\sqrt{\frac{\mathrm{CC}^\sigma(\lambda, \widehat{\mathcal{F}}, \mathcal{Z}_n^H)}{n}} \sum_{h=1}^H \beta^h + \frac{\mathrm{CC}^\sigma(\lambda, \widehat{\mathcal{F}}, \mathcal{Z}_n^H)}{n} \cdot \frac{\sum_{h=1}^H \beta^h}{\alpha}$$

$$\leq \sqrt{\frac{\mathrm{CC}^\sigma(\lambda, \widehat{\mathcal{F}}, \mathcal{Z}_n^H)}{n}} \cdot c_\beta \big( \alpha \zeta + H \sqrt{\ln N} \big) + \frac{\mathrm{CC}^\sigma(\lambda, \widehat{\mathcal{F}}, \mathcal{Z}_n^H)}{n} \cdot c_\beta \Big( \zeta + \frac{H \sqrt{\ln N}}{\alpha} \Big)$$

$$= \tilde{\mathcal{O}}\Big( \frac{3H \sqrt{\mathrm{CC}^\sigma(\lambda, \widehat{\mathcal{F}}, \mathcal{Z}_n^H) \cdot \ln N}}{\sqrt{n}} + \frac{3\zeta \cdot \mathrm{CC}^\sigma(\lambda, \widehat{\mathcal{F}}, \mathcal{Z}_n^H)}{n} \Big),$$

where we choose $\alpha = H\sqrt{\ln N}/\zeta$. By invoking Lemma C.2, we obtain that with probability at least $1 - 2\delta$,

$$\mathrm{SubOpt}(\hat{\pi}, x) = \tilde{\mathcal{O}}\Big( \frac{H(\mathrm{CC}(\lambda, \widehat{\mathcal{F}}, \mathcal{Z}_n^H))^{1/4} \cdot (\ln N_n(\gamma))^{1/2}}{n^{1/2} (C(\widehat{\mathcal{F}}, \mu))^{1/4}} + \frac{\zeta (\mathrm{CC}(\lambda, \widehat{\mathcal{F}}, \mathcal{Z}_n^H))^{1/2}}{n (C(\widehat{\mathcal{F}}, \mu))^{1/2}} \Big).$$

Ultimately, we conclude the proof. $\qquad\square$

# D  Implementation Details

**Implementation of UWMSG.**  Following prior uncertainty-based offline RL algorithm Model Standard-deviation Gradients (MSG) [16], we learn a group of $Q$ networks $Q_{w_i}, i = 1, \ldots, K$ with independent targets and optimize a policy $\pi_\theta$ with a lower-confidence bound (LCB) objective [16, 2]. Specifically, $Q_{w_i}$ is learned to minimize the following weighted regression objective with samples from the offline dataset $\mathcal{D}$:

$$\min_{w_i} \mathbb{E}_{(x,a,r,x') \sim \mathcal{D}} \left[ \frac{\left( \widehat{\mathcal{T}} Q_{w_i}(x,a) - Q_{w_i}(x,a) \right)^2}{\sigma(x,a)^2} \right]. \tag{43}$$

The weight function $\sigma$ is estimated via bootstrapped uncertainty: $\sigma(x,a) = \mathrm{clip}(\mu \times \sqrt{\mathbb{V}_{i=1,\ldots,K}[Q_{w_i}(x,a)]}, 1, M)$, where $\mathbb{V}_{i=1,\ldots,K}[Q_{w_i}]$ refers to the variance between the group of $Q$ functions, and $M$ is used to control the maximum value of the weighting function. Note that $\sigma(x,a)$ is detached from the gradients, and the update is exclusively on $w_i$. We introduce the uncertainty ratio $\mu$ for $\sigma(x,a)$ to tune the weight function. The independent target $\widehat{\mathcal{T}} Q_{w_i}$ for $Q_{w_i}$ is defined as follows:

$$\widehat{\mathcal{T}} Q_{w_i}(x,a) := r(x,a) + \gamma \mathbb{E}_{a' \sim \pi_\theta(\cdot|x')} \left[ Q_{w_i'}(x', a') \right], \tag{44}$$

where $Q_{w_i'}$ is the target network for $Q_{w_i}$. In empirical offline RL, it is a common practice [1, 2, 16] to utilize the discounted form of the Q function rather than the episodic version. The policy $\pi_\theta$ optimizes the same pessimistic objective as MSG:

$$\min_\theta \mathbb{E}_{x \sim \mathcal{D}, a \sim \pi_\theta(\cdot|x)} \left[ \mathbb{E}_{i=1,\ldots,K}[Q_{w_i}(x,a)] - \beta \cdot \sqrt{\mathbb{V}_{i=1,\ldots,K}[Q_{w_i}(x,a)]} \right], \tag{45}$$

Although the weighting function $\sigma(x,a)$ shares some similarities with the pessimistic bonus, they differ in the following aspects: (1) $\sigma(x,a)$ **measures the intrinsic variance of the corrupted data, while the pessimistic bonus penalizes out-of-distribution (OOD) actions produced by $\pi_\theta$; (2) $\sigma(x,a)$ weights the $Q$ learning objective and is detached from gradients, whereas the pessimistic bonus requires gradients for $\pi_\theta$.**

**Training and Evaluation Details.**  We use 3-layer MLPs with 256 neurons in each layer for both $Q$ and policy networks. The ensemble size is set to $K = 10$ for all the experiments. The hyperparameters, such as learning rate and optimizer, are listed in Table 2. We train each algorithm for 3000 epochs, where one epoch contains 1000 updates. Regarding the offline datasets, we use 'halfcheetah-medium-v2', 'walker2d-medium-replay-v2', and 'hopper-medium-replay-v2' datasets and refer to them as 'halfcheetah', 'walker2d', and 'hopper' in our paper. Our implementation is based on SAC-N [1]. Therefore, there are additional entropy regularization terms for both $\widehat{\mathcal{T}} Q_{w_i}(s,a)$ in Eq (44) and the policy objective in Eq (45). For hyperparameters related to corruption and uncertainty weighting, we list them in Table 3. Since tasks vary in their ability to resist corruption, their hyperparameters are tuned separately. We use the same LCB ratio $\beta$ for MSG and UWMSG, which is searched within $\{4.0, 6.0\}$. The uncertainty ratio $\mu$ for UWMSG is search from $\{0.2, 0.3, 0.5, 0.7, 1.0\}$.

To evaluate algorithms, we run the deterministic policy of each agent for 1000 steps and report their average cumulative returns with standard deviations over 10 random seeds. Our code is based on [42] and is available at https://github.com/YangRui2015/UWMSG.

**Data Corruption Details.**  We implement both random and adversarial corruption on either rewards or dynamics. The four types of data corruption are listed below:

- Random reward attack: randomly sample $c\%$ transitions $(x, a, r, x')$ from $D$, and modify the reward $\hat{r} \sim \mathrm{Uniform}[-\epsilon, \epsilon]$, where $c$ is the corruption rate and $\epsilon$ is the corruption scale.

- Random dynamics attack: randomly sample $c\%$ transitions $(x, a, r, x')$, and modify the next-step state $\hat{x}' = x' + \delta \cdot \mathrm{std}, \delta \sim \mathrm{Uniform}[-\epsilon, \epsilon]^d$, where $d$ is dimension of states and std is the $d$-dimensional standard deviation of all states in the offline dataset.

- Adversarial reward attack: randomly sample $c\%$ transitions $(x, a, r, x')$, and modify the reward as: $\hat{r} = -\epsilon \times r$.

- Adversarial dynamics attack: pretrain a group of $Q_p$ functions and a policy function $\pi_p$, then randomly sample $c\%$ transitions $(x, a, r, x')$, and modify the next-step states $\hat{x}' = \min_{\hat{x}' \in \mathbb{B}_d(x', \epsilon)} Q_p(x', \pi_p(\hat{x}'))$, where $\mathbb{B}_d(x', \epsilon) = \{|\hat{x}' - x'| \leq \epsilon \cdot \text{std}\}$ regularizes the maximum difference for each state dimension. The optimization is implemented through gradient descent similar to prior works [58, 53].

For the implementation of an adversarial dynamics attack, the optimization is performed through 10-step gradient descent with learning rate $\frac{\epsilon}{10}$. After each gradient descent step, the states are clipped within $\mathbb{B}_d(x', \epsilon)$. The pretraining algorithm used is MSG for the halfcheetah and walker2d tasks, while EDAC is employed for the hopper task due to its significantly better performance on this task compared to MSG in the absence of corruption. Finally, the corrupted data is saved and will be loaded for future training. To control the cumulative corruption $\zeta$ under continuous state-action spaces, we incorporate random or adversarial noise with predefined corruption ranges and corruption scales into the rewards and next-step states. This is because the fact that $\zeta_i \leq |r - r_\mathcal{D}| + D_{TV}(P\|P_\mathcal{D}) \leq |r - r_\mathcal{D}| + \sqrt{\frac{1}{2} D_{KL}(P\|P_\mathcal{D})}$. When we consider $P$ and $P_\mathcal{D}$ are both Diagonal Gaussian distributions with the same constant variance, $\zeta_i \leq |r - r_\mathcal{D}| + \sqrt{\frac{1}{2}}\|\mu - \mu_\mathcal{D}\| + \text{const}$. When we corrupt only one element in rewards and dynamics, the empirical cumulative corruption can be approximated as the multiplication of the number of corrupted samples and the corruption scale: $\zeta = |D| \times c\% \times \epsilon$, where $|D|$ represents the size of the dataset, and $\epsilon$ represents the corruption scale. Note that this approximation may not hold for our reward corruption, but we use the same calculation for simplicity.

Table 2: Hyper-parameters for UWMSG and MSG.

| Hyper-parameters | Value |
|---|---|
| Ensemble size $K$ | 10 |
| Policy network | FC(256,256,256) with ReLU |
| $Q$-network | FC(256,256,256) with ReLU |
| LCB ratio $\beta$ | $\{4.0, 6.0\}$ |
| Uncertainty ratio $\mu$ | $\{0.2, 0.3, 0.5, 0.7, 1.0\}$ |
| Maximum value of uncertainty weight $M$ | 10 |
| Target network smoothing coefficient $\tau$ | 5e-3 |
| Discount factor $\gamma$ | 0.99 |
| Policy learning rate | 3e-4 |
| $Q$ network learning rate | 3e-4 |
| Optimizer | Adam |
| Automatic Entropy Tuning | True |
| batch size | 256 |

Table 3: Data corruption settings and the hyperparameters used for uncertainty-weighting in UWMSG.

| Attack type | Attack object | Environment | Corruption rate $c\%$ | Corruption scale $\epsilon$ | Cumulative corruption $\zeta$ | LCB ratio $\beta$ | Uncertainty ratio $\mu$ |
|---|---|---|---|---|---|---|---|
| Random | Reward | halfcheetah | 20% | 30.0 | $5.99 \times 10^6$ | 4.0 | 0.7 |
| | | walker2d | 30% | 30.0 | $2.72 \times 10^6$ | 4.0 | 0.3 |
| | | hopper | 20% | 30.0 | $2.41 \times 10^6$ | 6.0 | 0.7 |
| | Dynamics | halfcheetah | 20% | 2.0 | $4.00 \times 10^5$ | 4.0 | 0.5 |
| | | walker2d | 10% | 0.5 | $1.51 \times 10^4$ | 6.0 | 0.5 |
| | | hopper | 10% | 0.5 | $2.01 \times 10^4$ | 6.0 | 0.7 |
| Adversarial | Reward | halfcheetah | 20% | 3.0 | $5.99 \times 10^5$ | 4.0 | 0.7 |
| | | walker2d | 20% | 3.0 | $1.81 \times 10^5$ | 4.0 | 0.5 |
| | | hopper | 10% | 5.0 | $2.01 \times 10^5$ | 6.0 | 0.7 |
| | Dynamics | halfcheetah | 30% | 1.2 | $3.60 \times 10^5$ | 4.0 | 0.2 |
| | | walker2d | 10% | 0.3 | $9.05 \times 10^3$ | 4.0 | 0.5 |
| | | hopper | 10% | 0.5 | $2.01 \times 10^4$ | 6.0 | 1.0 |

# E  Comparison with Uncertainty-weighted Actor Critic (UWAC)

Our practical implementation algorithm, UWMSG, shares some similarities with UWAC in terms of utilizing uncertainty weighting technique for offline RL. However, there are three key differences between our approaches:

1. We focus on offline RL with data corruption, rather than the general offline RL setting explored by UWAC. Therefore, in our setting, the uncertainty arises from both corrupted datasets and OOD actions.

2. While UWAC penalizes OOD actions in the Q objective through minimizing $\mathbb{E}_{(x,a,r,x')\sim\mathcal{D},a'\sim\pi_\theta(\cdot|x')}\left[\frac{\left(\widehat{\mathcal{T}}Q(x,a)-Q(x,a)\right)^2}{Var[Q(x',a')]}\right]$, with the aim of reducing the importance of OOD actions, our uncertainty weighting focuses on penalizing in-dataset $(x,a)$ pairs.

3. Another distinction lies in the uncertainty estimation methods employed. UWAC uses dropout uncertainty, while we utilize bootstrapped uncertainty in our work. However, it is worth noting that our approach is not limited to a specific type of uncertainty estimation. In the future, more advanced uncertainty estimation methods (e.g., [41, 7]) can be applied to potentially enhance the performance of UWMSG.

# F  Additional Results

**Learning Curves**  All learning curves are shown in Figure 2, Figure 3, and Figure 4. We can find that (1) current offline RL methods are susceptible to data corruption, e.g., MSG, EDAC, SAC-N achieve poor and unstable performance under adversarial attacks, and (2) our proposed UWMSG method significantly improves performance under different data corruption scenarios. Moreover, we posit that the reason for the observed initial increase and subsequent significant decrease of EDAC and SAC-N performance in some cases may be attributed to the characteristics of Temporal Difference (TD) learning. Specifically, the effect of corruption needs to accumulate over time, which may necessitate an extended training period to destroy the performance. In contrast, our algorithm UWMSG does not suffer from this problem and exhibits stable performance.

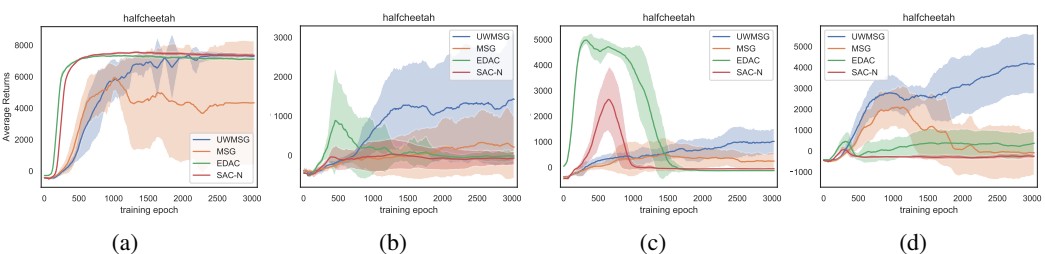

Figure 2: Comparison on the halfcheetah task under (a) random reward, (b) random dynamics, (c) adversarial reward, and (d) adversarial dynamics attacks.

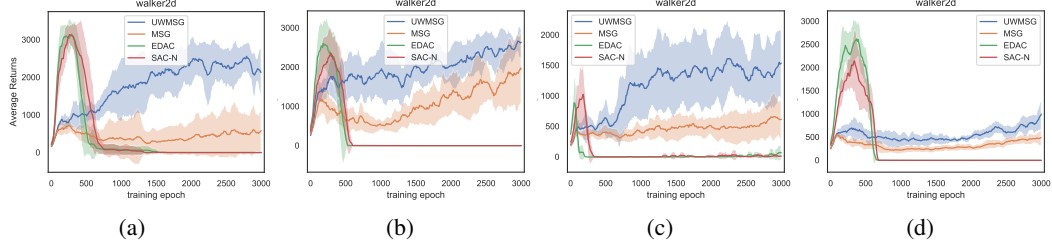

Figure 3: Comparison on the walker2d task under (a) random reward, (b) random dynamics, (c) adversarial reward, and (d) adversarial dynamics attacks.

**Varying Corruption Level**  We evaluate the performance of UWMSG under varying levels of corruption in Figure 5. This is achieved by maintaining a consistent corruption scale in Table 3 while adjusting the corruption rate. As depicted in the figure, as the cumulative corruption level rises, the overall performance of UWMSG progressively declines. These findings align with our theoretical analysis. Besides, the results indicate that dynamics corruption poses a greater challenge compared to reward corruption, leading to a larger drop in performance with a smaller corruption level.

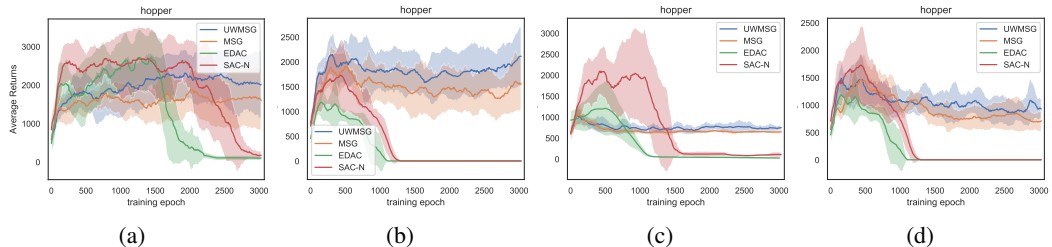

(a)          (b)          (c)          (d)

Figure 4: Comparison on the hopper task under (a) random reward, (b) random dynamics, (c) adversarial reward, and (d) adversarial dynamics attacks.

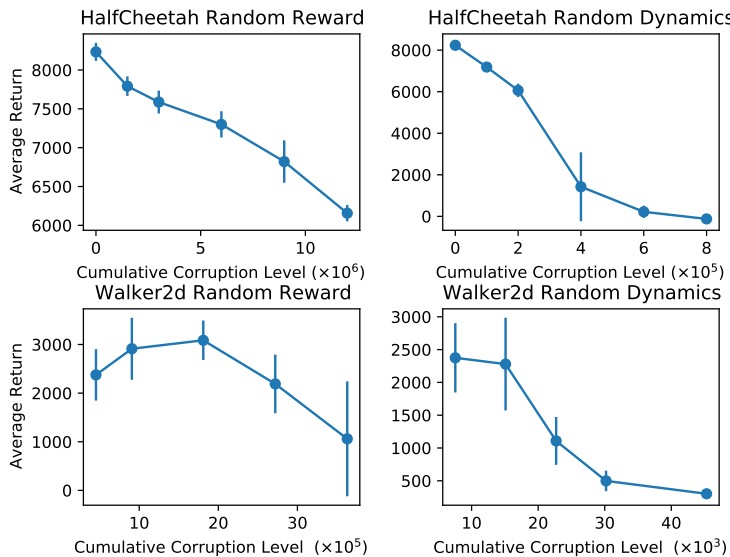

Figure 5: Performance of UWMSG under varying levels of corruption. Results are averaged over 5 random seeds.

## G    Technical Lemmas

**Lemma G.1** (Lemma 3.1 of Jin et al. [24])**.** *Let* $\hat{\pi} = \{\hat{\pi}^h\}_{h=1}^H$ *be the greedy policy such that for any* $x$, $\hat{\pi}^h(x) = \operatorname{argmax}_{a \in \mathcal{A}} f_n^h(x, a)$. *For any initial state* $x \in \mathcal{X}$,

$$\operatorname{SubOpt}(\hat{\pi}, x) = \sum_{h=1}^H \mathbb{E}_{\pi_*}\left[f_n^h(x^h, \pi_*(x^h)) - f_n^h(x^h, \hat{\pi}(x^h)) \,\middle|\, x^1 = x\right]$$

$$- \sum_{h=1}^H \mathbb{E}_{\pi_*}\left[\mathcal{E}^h(f_n, x^h, a^h) \,\middle|\, x^1 = x\right] + \sum_{h=1}^H \mathbb{E}_{\hat{\pi}}\left[\mathcal{E}^h(f_n, x^h, a^h) \,\middle|\, x^1 = x\right],$$

*where* $\mathcal{E}^h(f, x^h, a^h) = f^h(x^h, a^h) - (\mathcal{T}^h f^{h+1})(x^h, a^h)$ *is the Bellman residual.*

**Lemma G.2.** *Let* $\{\epsilon_s\}$ *be a sequence of zero-mean conditional* $\eta$-*sub-Gaussian random variables:* $\ln \mathbb{E}[e^{\lambda \epsilon_s}|\mathcal{S}_{s-1}] \leq \lambda^2 \eta^2/2$, *where* $\mathcal{S}_{s-1}$ *represents the history data. We have for* $t \geq 1$, *with probability at least* $1 - \delta$,

$$\sum_{s=1}^t \epsilon_i^2 \leq 2t\sigma^2 + 3\sigma^2 \ln(1/\delta).$$

*Proof.* The proof can is presented in Lemma G.2 of Ye et al. [55].     □

**Lemma G.3** (Lemma G.4 of Ye et al. [55]). *Consider a function space $\mathcal{F} : \mathcal{Z} \to \mathbb{R}$ and filtered sequence $\{z_t, \epsilon_t\}$ in $\mathcal{X} \times \mathbb{R}$ so that $\epsilon_t$ is conditional zero-mean $\eta$-sub-Gaussian noise. For $f_*(\cdot) :$ $\mathcal{Z} \to \mathbb{R}$, suppose that $y_t = f_*(z_t) + \epsilon_t$ and there exists a function $f_b \in \mathcal{F}$ such that for any $t \in [T]$, $\sum_{s=1}^{t} |f_*(z_s) - f_b(z_s)| := \sum_{s=1}^{t} \zeta_s \leq \zeta$. If $\hat{f}_t$ is an (approximate) ERM solution for some $\epsilon' \geq 0$:*

$$\left( \sum_{s=1}^{t} (\hat{f}_t(z_s) - y_s)^2 / \sigma_s^2 \right)^{1/2} \leq \min_{f \in \mathcal{F}_{t-1}} \left( \sum_{s=1}^{t} (f(z_s) - y_s)^2 / \sigma_s^2 \right)^{1/2} + \sqrt{t} \epsilon',$$

*with probability at least $1 - \delta$, we have for all $t \in [T]$:*

$$\sum_{s=1}^{t} (\hat{f}_t(z_s) - f_b(z_s))^2 / \sigma_s^2 \leq 10\eta^2 \ln(2N(\gamma, \mathcal{F}, \|\cdot\|_\infty)/\delta) + 5 \sum_{s=1}^{t} |\hat{f}_t(z_s) - f_b(z_s)| \zeta_s / \sigma_s^2$$
$$+ 10(\gamma + \epsilon')\big((\gamma + \epsilon')t + \sqrt{t C_1(t, \zeta)}\big),$$

*where $C_1(t, \zeta) = 2(\zeta^2 + 2t\eta^2 + 3\eta^2 \ln(2/\delta))$.*

*Proof.* The proof can be seen in Lemma G.4 of Ye et al. [55]. $\qquad\square$

**Lemma G.4** (Theorem 1.3 of Tropp [43]). *For a finite sequence $\{X_i\}_{i \in [n]}$ of independent, random and self-adjoint matrices with dimension $d$, let $\{A_i\}_{i \in [n]}$ be a sequence of fixed self-adjoint matrices. If each random matrix satisfies*

$$\mathbb{E} X_i = 0 \quad X_i^2 \preceq A_i^2 \quad \text{almost surely,}$$

*then, for all $t \geq 0$,*

$$\mathbb{P}\Big( \lambda_{\max}\big( \sum_{i=1}^{n} X_i \big) \geq t \Big) \leq d \cdot e^{-t^2/(8\sigma^2)},$$

*where $\sigma^2 = \| \sum_{i=1}^{n} A_i^2 \|_{\mathrm{op}}$.*