# OpenReview forum: "Corruption-Robust Offline Reinforcement Learning with General Function Approximation"
_NeurIPS.cc/2023/Conference — NeurIPS 2023 poster_

### Official Review · Reviewer_wcW2 · 2023-06-22

**Soundness:** 3 good
**Presentation:** 4 excellent
**Contribution:** 3 good
**Rating:** 6
**Confidence:** 3

**Summary:**

The paper studies offline RL with general function approximation in the presence of adversarial corruption. Specifically, an adversary can corrupt the rewards and transitions in the offline dataset with a fixed cumulative budget. The corruption formulation encompasses prior corruption formulations. The proposed algorithm is based on an uncertainty-weight iteration subroutine, inspired by similar techniques in prior robust online RL papers. The authors conduct a finite-sample analysis, showing a tighter result on the corruption term. The theory is accompanied by a practical implementation tested over a couple of D4RL tasks.

**Strengths:**

- The authors propose a corruption formulation, which admits prior formulations such as fixed fraction contamination of Zhang et al. as special cases.
- The theoretical result improves over the most relevant prior work Zhang et al. from $\sqrt{\epsilon}$ dependency to $\epsilon$ dependency of the corruption term, which matches the lower bound when specialized to the linear case.
- Inspired by the uncertainty weighting approach in robust online RL, the authors propose a similar weight iteration method in the offline RL case. However, in the offline case, the
- The authors implement their algorithm over a subset of continuous control tasks in D4RL.

**Weaknesses:**

- The main limitation of this work is that the proposed algorithm requires the corruption level to be known. I did not find any attempts on possible heuristics for estimating the corruption level, which limits the practicality of the proposed approach.
- The experiments section is generally weak. The practical implementation is limted to only two continous control tasks (Walker and Half Cheetah) from D4RL that are artificially corrupted. It would be good to see and compare performance with other tasks in D4RL as well. Furthermore, the approach is compared with uncertainty-based offline RL algorithms that are not designed to be corruption robust, making the comparison unfair.

**Questions:**

Suggestions:
- Perhaps comparison with an implementation of Zhang et al instead of uncertainty-based offline RL is more appropriate.

**Limitations:**

Yes, clearly stated.

---

> ### Author Rebuttal · Authors · 2023-08-10
>
> Thank you for your positive review and constructive evaluation!
>
> **Q1** The main limitation of this work is that the proposed algorithm requires the corruption level to be known. I did not find any attempts at possible heuristics for estimating the corruption level, which limits the practicality of the proposed approach.
>
> **A1**
> - In the simulation, we can treat the uncertainty ratio $\alpha=O(1/\zeta)$ as a hyperparameter that can be tuned during the experimentation phase. The use of independent and identically distributed (i.i.d.) trajectories in our experiments facilitates the adjustment of this parameter, rendering the tuning process straightforward and conducive to optimize the performance.
> - We can also offer a choice of $\alpha$, which is $\Theta(1/\sqrt{n})$. This choice finds support in the online setting, as demonstrated in [1], where this specific $\alpha$ configuration ensures that suboptimality remains aligned with uncorrupted conditions, even when $\zeta=O(\sqrt{n})$.
>
> ---
>
> **Q2** The experiments section is generally weak. The practical implementation is limited to only two continuous control tasks (Walker and Half Cheetah) from D4RL that are artificially corrupted. It would be good to see and compare performance with other tasks in D4RL as well.
>
> **A2**
> - Thanks for your suggestion. Since we consider 4 types of training-time attacks for each task, the computational burden is much higher compared to typical empirical offline RL works. However, we agree that incorporating more tasks would strengthen the empirical validation of our approach.
> - To solve your concern, during the rebuttal period, we have conducted additional experiments on the Hopper task (another D4RL task). The results from these experiments consistently align with our empirical findings in the paper, demonstrating that EDAC and SACN struggle to handle corrupted data, while our empirical algorithm UWMSG achieves robust performance. More results will be provided in the revision.
>
> |  Hopper  | UWMSG                         | MSG            | EDAC                            |  SACN
> |---:|:-------------------|:-------------------|:------------------|:-------------------|
> |  Hopper random reward | **2481.21 $\pm$ 866.62** | 1488.48 $\pm$ 522.98 | 77.74 $\pm$ 31.13 | 225.65 $\pm$ 138.14 |
> |  Hopper random dynamics | **1511.44 $\pm$ 521.08** | 951.82 $\pm$ 187.02 | 5.79 $\pm$ 1.38 | 3.37 $\pm$ 2.3 |
>
> ---
>
> **Q3** The approach is compared with uncertainty-based offline RL algorithms that are not designed to be corruption robust, making the comparison unfair.
>
> **A3** To our knowledge, there is very little empirical work on corruption-robust offline RL. Zhang et al. assume that there exists a robust regression oracle, but they didn't give specific implementation details about the oracle. We will try to find a corruption-robust estimation method to implement Zhang et al. or other corruption-robust algorithms in the revision.
>
> [1] Ye et al. Corruption-robust algorithms with uncertainty weighting for nonlinear contextual bandits and Markov decision processes.
>
> [2] Zhang et al. Corruption-robust offline reinforcement learning.

---

### Official Review · Reviewer_GirP · 2023-06-26

**Soundness:** 3 good
**Presentation:** 4 excellent
**Contribution:** 3 good
**Rating:** 6
**Confidence:** 3

**Summary:**

The research investigates the issue of corruption robustness in offline reinforcement learning with general function approximation, where an adversary can corrupt samples in the offline dataset. The objective is to find a policy that remains robust against such corruption and minimizes the suboptimality gap compared to the optimal policy in uncorrupted Markov decision processes. To address this, a new uncertainty weight iteration procedure inspired by the robust online RL setting is designed, enabling efficient computation on batched samples. The proposed corruption-robust algorithm achieves a suboptimality bound, considering single policy coverage and knowledge of the corruption level. The analysis reveals that the corruption-dependent error term matches the existing lower bound for corrupted linear MDPs, indicating the tightness of the analysis in terms of corruption dependence.

**Strengths:**

I think this work really pushes the data-corruption solving real-world RL community research efforts further by answering:

> Can we design a generic algorithm for data-corrupted offline RL in the context of function approximation?

The main contribution: algorithmic introduction to uncertainty reweighting idea. Both theoretical and practical performances have been shown. This is good for publication at NeurIPS.

**Weaknesses:**

I have only one minor weakness for this work as follows:

On Assumption 2.1. The recent offline RL theoretical guarantees proposing pessimism is to avoid completeness assumption and having weaker concentrabilities, like that of [23]. I don't see what is pessimism helping in this work if not to avoid the completeness. Maybe I am missing something. I am okay with this assumption if the work is trying to address the corruption-robust problem which requires (?) completeness?

**Questions:**

please see weaknesses for more.

**Limitations:**

Limitations is marked Yes, but I couldn't find the discussion in the main paper.

---

> ### Author Rebuttal · Authors · 2023-08-10
>
> Thank you for your positive feedback and valuable comments!
>
> **Q1** On Assumption 2.1. I don't see what is pessimism helping in this work if not to avoid completeness. I am okay with this assumption if the work is trying to address the corruption-robust problem which requires (?) completeness?
>
> **A1**
> - Pessimism is used to relax the coverage condition (e.g., concentrability condition) rather than completeness. Completeness is for general function approximation.  Specifically, in the domain of linear MDPs [1,2,3], where the value functions exhibit a linear structure, completeness holds trivially. Nevertheless, pessimism remains necessary even in this linear framework. Without pessimism, a uniform coverage assumption becomes necessary, as demonstrated by Theorem 3.2 in [4]. In contrast, leveraging pessimism enables us to rely on a less restrictive partial coverage, as demonstrated by Corollary 4.5 in [1] and Theorem 3.3 in [4].
> - In general function approximation, completeness plays a pivotal role, ensuring that the Bellman backup of the value function resides within the confines of the considered function space. In [3], they also assume completeness and use pessimism.
> - In conclusion, completeness is used for general function approximation. Pessimism is established for offline RL without a uniform coverage condition. They lack direct connections.
>
> ---
>
> **Q2** Limitations are marked Yes, but I couldn't find the discussion in the main paper.
>
> **A2** We discuss the limitation in Lines 318-323, which are left as future works.
>
>
> [1] Jin, Y., Yang, Z., and Wang, Z. (2021b). Is pessimism provably efficient for offline rl?
>
> [2] Xiong et al. Nearly minimax optimal offline reinforcement learning with linear function approximation: Single-agent mdp and Markov game.
>
> [3] Xie et al. Bellman-consistent pessimism for offline reinforcement learning
>
> [4] Zhang et al. Corruption-robust offline reinforcement learning.

---

> > ### Comment · Reviewer_GirP · 2023-08-20
> >
> > The rebuttal addressed my concerns. I have adjusted my rating considering the rebuttal and other reviewers’ concerns.

---

> > > ### Author Response · Authors · 2023-08-21
> > >
> > > We're grateful for your positive feedback. We are pleased to learn that our response has effectively addressed the concerns raised. We've noticed that the current ratings are still unchanged. We kindly remind the reviewer to adjust the rating if the reviewer forgot.

---

### Official Review · Reviewer_p9Rd · 2023-07-05

**Soundness:** 4 excellent
**Presentation:** 3 good
**Contribution:** 4 excellent
**Rating:** 7
**Confidence:** 4

**Summary:**

This paper proposed a new robust offline RL algorithm called the Corruption Robust PEVI (CR-PEVI). The CR-PEVI achieves smaller suboptimality error compared to existing methods. The key idea behind CR-PEVI is to utilize the uncertainty-weighting technique when solving the least-squares problem in PEVI. The weights in the proposed method do not have closed-form solution, thus the authors proposed an iterative approach and relies on the monotone convergence theorem to prove convergence to the target weights. Based on that and several other assumptions, e.g., dataset needs to be well-explored, the paper proved an upper bound on the suboptimality of the policy learned by the CR-PEVI algorithm. When specializing to the linear function approximation with dimension d, the proposed CR-PEVI achieves O(eta*d/n) convergence rate. Experiments demonstrate the effectiveness of the CR-PEVI method.

**Strengths:**

(1). This paper studied robust offline RL, which is a hot and timely topic in recent years. Prior works mostly consider test-time attacks that are relatively easier to defend. However, less is understood about robustness against training-time attacks. This paper pushes the frontier in this domain and gained deeper understanding on how to design poisoning-resilient offline RL algorithms in the function approximation scenario, which is important and fascinating work.

(2). The theoretical analysis is intriguing. The authors combined the uncertainty-weighting technique and the PEVI algorithm to propose a new algorithm that is robust to offline poisoning attacks. The analysis is based on several critical theoretical results established in the paper, which are complete and solid. Overall, the paper has strong theory contributions.

(3). The empirical results are also convincing. On all the tasks, the CR-PEVI achieves much better performance compared to baselines.

**Weaknesses:**

(1). It seems like the definition of cumulative corruption is not commonly used by prior works. It is defined as the gap between the Bellman operators induced by the clean dataset and corrupted dataset, which is less intuitive and harder to quantize. In contrast, it makes more sense to define corruption level as the real modification on the reward/state transition over the trajectories.

(2). In the experiment, the setup does not strictly conform with attacks under limited budget. The authors implemented different attacks without reporting the corruption level for every different attack and victim algorithm.  Furthermore, there is not parameter study on the corruption level eta as well. This makes me wonder if the comparison to prior works, in particular, Zhang et al, is fair enough.

(3) Missing a few related references on training-time attacks in RL:

[1]. Policy Poisoning in Batch Reinforcement Learning and Control

[2]. Adaptive Reward-Poisoning Attacks against Reinforcement Learning

**Questions:**

(1). Why the paper chooses to define the cumulative corruption based on the Bellman operator instead of the modification on the dataset? Is the goal to favor theoretical analysis? How does the theory changes if the corruption level is defined differently, e.g., total change on the rewards?

(2). It's important to report the corruption level induced by different attack and victim RL algorithms in the experiment section. Furthermore, I was wondering how the cumulative corruption level affects the policy learned by CR-PEVI in practical scenarios.

**Limitations:**

Yes

---

> ### Author Rebuttal · Authors · 2023-08-10
>
> Thank you for your positive feedback and suggestions!
>
> **Q1** The definition of cumulative corruption is less intuitive and harder to quantize. Is the goal to favor theoretical analysis? It makes more sense to define corruption level as the real modification of the reward/state transition over the trajectories. How does the theory change if the corruption level is the total change in rewards?
>
> **A1**
> - The definition of cumulative corruption is the same as that in the online RL literature [1,2].
> - This definition is more general and concise, which contributes to a sense of uniformity and clarity within our analysis, as it avoids segregating the analysis of corruption effects on rewards and transition probabilities. In particular, since
> $$
> (\mathcal T^h g)(x,a) = r^h(x,a) + (\mathbb P^h g^{h+1})(x,a),
> $$
> the modification on $r^h$ or $\mathbb P^h$ contributes to the modification on $\mathcal T^h$.
> - In our setting, during the offline dataset collection, the agent assembles complete trajectories with a horizon length $H$, while an adversary systematically corrupts $r^h$ and $x^{h+1}$ at each step $h$ before they are unveiled to the agent. Measuring the real modification on each tuple $(s,a,s',r)$ is hard in our setting since once one modifies a tuple, the subsequent trajectory shifts. It is difficult to tell whether the shift is due to corruption or the change from a previous step. Actually, we care about the part of the shift that violates the Bellman completeness.
> - Our corruption setting differs from [3], where each tuple can be corrupted independently. As a result, our framework does not supersede the results of [3]. We will restate this point in the revision and consider the corruption in this alternative setting as a future work.
> - The theory will not change if the corruption level is defined as the total change in the rewards because reward change is only a special case of our definition.
>
> ---
>
> **Q2** In the experiment, the setup does not strictly conform with attacks under a limited budget. There is no parameter study on the corruption level eta. If the comparison to prior works, in particular, [3], is fair enough?
>
> **A2**
> - We would like to clarify that the attacks in our experiments are under a limited budget and the comparison is fair. To control the cumulative corruption level, we use corruption rate $c\%$ and corruption scale $\epsilon$ as the fact that  $\zeta_i \leq |r-r_{\mathcal{D}}| + D_{TV}(P\\|P_{\mathcal{D}}) \leq |r-r_{\mathcal{D}}| + \sqrt{\frac{1}{2} D_{KL}(P\\|P_{\mathcal{D}}})$. When we consider $P$ and $P_{\mathcal{D}}$ are both diagonal Gaussian distributions with the same constant variance, $\zeta_i \leq |r-r_{\mathcal{D}}| + \sqrt{\frac{1}{2}} \|\mu - \mu_{\mathcal{D}}\|= \epsilon$. Then the cumulative corruption level $\zeta = |D| \times c\% \times \epsilon$, where $|D|$ is the size of the dataset.
> - The cumulative corruption levels for each attack in our experiments are $5.99 \times 10^6$, $2.72\times 10^6$, $4.00 \times 10^5$, $1.51 \times 10^4$, $5.99 \times 10^5$, $1.81\times 10^5$, $3.60 \times 10^5$, $9.05 \times 10^3$, according to the order in Table 3 (Appendix D). **We applied the same corruption level for every victim under each attack, so the comparison is  fair**. In **A4**, we provide further parameter study about the cumulative corruption level $\zeta$.
> - Regarding [3], it is a pure theoretical work without giving implementation details about the robust least-square oracle. As a result, we are not able to compare with it.
>
> ---
>
> **Q3** Missing related references on training-time attacks in RL
>
> **A3** Thanks for the suggestion. We will add them in the revision.
>
> ---
>
> **Q4** How the cumulative corruption level affects the policy learned by CR-PEVI in practical scenarios.
>
> **A4** In the following tables, the performance of CR-PEVI generally decreases with the increase in cumulative corruption level $\zeta$. However, it is worth noting that CR-PEVI exhibits a remarkable ability to handle a certain level of corruption, resulting in only a small performance loss, e.g., in the cases of Halfcheetah random reward and Walker2d random reward.
>
> |$\zeta$| 0.00|1.50 $\times 10^6$|3.00 $\times 10^6$|5.99 $\times 10^6$|8.99$\times 10^6$|1.20$\times 10^7$|
> |-:|:-|:-|:-|:-|:-|:-|
> |Halfcheetah random reward|8233.27$\pm$116.27|7791.98$\pm$125.79|7587.58$\pm$147.51|7287.53$\pm$157.53|6821.02$\pm$272.65|6156.51$\pm$105.38|
>
> |$\zeta$| 0.00|1.00$\times10^5$|2.00$\times10^5$|4.00$\times10^5$|6.00$\times10^5$|8.00$\times10^5$|
> |-:|:-|:-|:-|:-|:-|:-|
> |  Halfcheetah random dynamics | 8132.66 $\pm$ 102.65 | 7184.59 $\pm$ 209.46 | 6063.33 $\pm$ 320.4 | 1477.29 $\pm$ 519.26 | 219.32 $\pm$ 293.97 | -123.52 $\pm$ 117.3 |
>
> |$\zeta$|4.53$\times 10^5$| 9.07$\times 10^5$|1.81$\times10^6$|2.72$\times10^6$|3.63$\times10^6$|
> |---:|:-------------------|:-------------------|:------------------|:-------------------|:------------------|
> |Walker2d random reward|2375.03$\pm$528.4|2912.6$\pm$637.09|3086.47$\pm$406.06|3181.08$\pm$114.53|1060.2$\pm$1181.39|
>
> |$\zeta$|7.55$\times10^3$|1.51$\times10^4$|2.27$\times10^4$|3.02$\times10^4$|4.53$\times10^4$
> |---:|:-------------------|:-------------------|:------------------|:-------------------|:------------------|
> |Walker2d random dynamics|2154.99$\pm$797.72|2063.97$\pm$851.61|1107.42$\pm$364.81|498.34$\pm$156.48|302.09$\pm$38.13|
>
> [1] Ye et al. Corruption-robust algorithms with uncertainty weighting for nonlinear contextual bandits and Markov decision processes.
>
> [2] Wei et al. A model selection approach for corruption robust reinforcement learning.
>
> [3] Zhang et al. Corruption-robust offline reinforcement learning.

---

> > ### Comment · Reviewer_p9Rd · 2023-08-20
> >
> > Thank you for your response. My questions are carefully addressed in the rebuttal. I have increased my score accordingly.
> >
> > There are a few related works about poisoning attacks in RL, and the authors should consider discussing them in the paper.
> >
> > Ma et al. Policy poisoning in batch reinforcement learning and control (NeurIPS 19).
> >
> > Zhang et al. Adaptive Reward-Poisoning Attacks against Reinforcement Learning (ICML 20).

---

> > > ### Author Response · Authors · 2023-08-21
> > >
> > > Thank you for increasing the score! We will be sure to discuss the additional related works pointed out by the reviewer in the final version.

---

### Official Review · Reviewer_H58a · 2023-07-07

**Soundness:** 3 good
**Presentation:** 3 good
**Contribution:** 2 fair
**Rating:** 6
**Confidence:** 2

**Summary:**

This work considers corrupted offline RL with general function approximation, and propose an algorithm based on PEVI and the uncertainty weighting technique. Under certain date coverage and well-explored dataset assumptions, the proposed algorithm achieves neal-optimal bound, where the corruption-independent term matches with the uncorrupted bound while the corruption-dependent term nearly matches the lower bound for linear models. Futher, experiments showcase the efficetiveness of the algorithm.

**Strengths:**

1 This work is well-organized and well-written.

2 The studied corrupted offline RL with general function approximation is not being widely explored.

3 Adapting uncertainty-related weights technique to offline setting may be of interest.

**Weaknesses:**

1 The uncertainty-weighting technique and the PEVI are not new.

2 This work extends the corrupted online setting to corrupted offline setting


**Questions:**

1 Is the result comparable to the existing corrupted online RL with general function approximation?

2 Could the authors comment more on Eqn. (6) and give intuition? It is unclear what tuning parameter $\alpha$ and $\lambda$ are designed for in this uncertainty-related weights.

3 For the lower bound shown in Theorem 2, it would be great if the authors provide any insight.

In line 12, ''$\widehat{F}$'' -> (or <-) ''$\hat{F}$''.

---

> ### Author Rebuttal · Authors · 2023-08-10
>
> Thank you for your positive feedback and valuable comments!
>
> **Q1** The uncertainty-weighting technique and PEVI are not new. This work extends corrupted online setting to offline setting.
>
> **A1** Our work is not a trivial extension from online RL to offline RL. We would like to highlight our technical novelties in two main aspects.
> - While PEVI has been introduced for offline learning of linear MDPs, our innovation lies in its extension to the eluder-type general function approximation. To achieve this, we introduce a new partial coverage coefficient in Definition 4.1 and assume that the coefficient is finite, which implies the single-policy coverage condition for offline linear MDPs (Lines 225-232).
> - Second, our uncertainty-weighting techniques substantially differ from the online RL setting in algorithm and theoretical analysis.
>     - For the algorithm, we propose the Uncertainty Value Iteration algorithm in Algorithm 1 to address the challenges posed by the inability of offline algorithms to iterate over timesteps like online counterparts. Furthermore, in Lemma 3.1, we prove the convergence of this iteration using the monotone convergence theorem, and an approximation of the uncertainty quantity is sufficient for robust estimation.
>     - For the analysis, we develop a novel technique to eliminate the dependence of the suboptimality on the uncertainty-related weights. Specifically, the original suboptimality bound contains the weighted coverage coefficient $\text{CC}^{\sigma}(\lambda,\hat{\mathcal F},\mathcal Z_n^H)$ defined in Eqn. (13), which makes the bound instance-dependent and hard to compare with existing results. Unlike the online setting where one can assume the bounded eluder dimension and try to bound the sum of bonus by the eluder dimension, we demonstrate in Lemma 4.1 that if the dataset is well-explored (Assumption 4.1),
> $$
> \text{CC}^{\sigma}(\lambda,\hat{\mathcal F},\mathcal Z_n^H) = \tilde O\Big(\sqrt{\text{CC}(\lambda,\hat{\mathcal F},\mathcal Z_n^H)/C(\hat{\mathcal F},\mu)}\Big),
> $$
> where $\text{CC}(\lambda,\hat{\mathcal F},\mathcal Z_n^H)$ is the (unweighted) coverage coefficient, and $C(\hat{\mathcal F},\mu)$ is a constant measuring the complexity of the function space $\hat{\mathcal F}$. To obtain this conclusion, we use the uncertainty structure in the weights, decompose the coefficient into the product of the coverage term and density term, and apply the Cauchy-Schwarz inequality.
>
> ---
>
> **Q2** Is the result comparable to the existing corrupted online RL with general function approximation?
>
> **A2** Since online and offline RL adopt distinct assumptions, such as the bounded eluder dimension and coverage condition, their results are not directly comparable. Here we compare their results in a less formal manner.  Regardless of their different assumptions, our offline RL result aligns with the online RL result in terms of sample complexity. Specifically, our work establishes that achieving an $\epsilon$-suboptimality in the offline setting requires a number of samples that is at most $\tilde O(\epsilon^{-2}+\zeta\cdot\epsilon^{-1})$. As a comparison, the $\tilde O(\sqrt{n} + \zeta)$ regret proved in [1] also implies an $\tilde O(\epsilon^{-2}+\zeta\cdot\epsilon^{-1})$ sample complexity.
>
> ---
>
> **Q3** Could the authors comment more on Eqn. (6) and give intuition?
>
> **A3** The uncertainty quantity is a ratio between the prediction error and the training error, which depicts the diversity of a sample $z_i$ against the whole training samples $\{z_1,\ldots,z_n\}$. If the uncertainty quantity is larger than $\alpha$,
> $
> \sigma_i^2 = 1/\alpha\cdot \text{Uncertainty};
> $
> else, $\sigma_i=1$.
> - We also use linear function class to give intuition. When the function space $\mathcal F^h$ is embedded into a $d$-dimensional vector space: $\mathcal F^h=\{\langle w(f), \phi(\cdot) \rangle : z\rightarrow\mathcal R\}$. Then, the uncertainty quantity becomes
> $$
> \sup_{f,f'\in\mathcal F} \frac{|\langle w(f)-w(f'), \phi(z_i) \rangle|}{\sqrt{\lambda+\sum_{j=1}^n\big(\langle w(f)-w(f'), \phi(z_j) \rangle\big)^2/\sigma_j^2}} \le \sup_{f,f'\in\mathcal F} \frac{|\langle w(f)-w(f'), \phi(z_i) \rangle|}{\sqrt{\big(w(f)-w(f')\big)^{\top} \Lambda \big(w(f)-w(f')\big)}}
> \le \sqrt{\phi^{\top}(z_i)\Lambda^{-1}\phi(z_i)},
> $$
> where $\Lambda=\sum_{j=1}^n\phi(z_j)\phi^{\top}(z_j)/\sigma_j^2$. The $\sqrt{\phi^{\top}(z_i)\Lambda^{-1}\phi(z_i)}$ is the bonus/uncertainty in the linear setting. $\big(\phi^{\top}(z_i)\Lambda^{-1}\phi(z_i)\big)^{-1}$ represents the effective number of samples in the $\{z_i\}_{i=1}^n$ along the $z_i$ direction. Moreover, we discuss in Lemma B.3 that under mild conditions the linear and nonlinear uncertainty quantities are almost equivalent.
>
> ---
>
> **Q4** What tuning parameter is designed for $\alpha$, $\lambda$ in this uncertainty-related weights.
>
> **A4** In Theorem 1, we choose $\alpha=H\sqrt{\ln N_n(\gamma)}/\zeta$, $\lambda=\ln(N_n(\gamma))$, where $H$ is the horizon of the MDP, $N_n$ is the covering number, and $\zeta$ is the cumulative corruption level. We can add a multiplicative constant to $\alpha$ and $\lambda$ respectively, and tune the constant in practice.
>
> ---
>
> **Q5** Provide insight the lower bound in Theorem 2.
>
> **A5** Following [2], we consider the tabular MDP with $d=SA$ and corruption on the reward. By the pigeonhole principle, there must exist a sample $(s,a)$ with the smallest empirical distribution $\nu(s,a)\le 1/SA$.  Subsequently, an adversary strategically introduces corruption to all instances of $(s,a)$, thus modifying the reward to $\tilde r(s,a)=r(s,a)+SA\zeta/(nH)$. Then, a confusing pair $(s',a')$ satisfying that $r(s',a')=r(s,a)+SA\zeta/(2nH)$ will make it hard to distinguish $(s,a)$ and $(s',a')$, leading to at least $SA\zeta/(2n)$ suboplimality gap.
>
> [1] Ye et al. Corruption-robust algorithms with uncertainty weighting for nonlinear contextual bandits and Markov decision processes.
>
> [2] Zhang et al. Corruption-robust offline reinforcement learning.

---

> > ### Comment · Reviewer_H58a · 2023-08-15
> >
> > Thanks for your rebuttal. Your clarifications have addressed my concern amd I raises rating from 5 to 6.

---

> > > ### Author Response · Authors · 2023-08-15
> > >
> > > Thank you for raising the score. We're glad to know that our response has addressed your concerns. We will incorporate these clarifications into the final version.

---

### Author Rebuttal · Authors · 2023-08-10

We express our gratitude to the reviewers for their positive feedback and constructive suggestions!

During the rebuttal period, we make some explanations and illustrations for our theory. For experiments, we complement two experiments and present the results in the following pdf. Firstly, we show the performance of our algorithm under different cumulative corruption levels in Figure 1. Secondly, we conduct additional experiments on the Hopper task in Table 1. We will add more empirical results in the revision according to the suggestions of reviewers.

---

### Decision · Program_Chairs · 2023-09-21

**Decision:**

Accept (poster)

**Comment:**

After the discussion, the reviewers agree that that paper provides a new contribution to the offline corruption robust RL by improving the previous work's bound so that it matches the lower bound now. The reviewers agree that the paper should be accepted.